# TeZO: Empowering the Low-Rankness on the Temporal Dimension in the Zeroth-Order Optimization for Fine-tuning LLMs

## Abstract

Zeroth-order optimization (ZO) has demonstrated remarkable promise in efficient fine-tuning tasks for Large Language Models (LLMs). In particular, recent advances incorporate the low-rankness of gradients, introducing low-rank ZO estimators to further reduce GPU memory consumption. However, most existing works focus solely on the low-rankness of each individual gradient, overlooking a broader property shared by all gradients throughout the training, i.e., all gradients approximately reside within a similar subspace. In this paper, we consider two factors together and propose a novel low-rank ZO estimator, `TeZO`, which captures the low-rankness across both the model and temporal dimension. Specifically, we represent ZO perturbations along the temporal dimension as a 3D tensor and employ Canonical Polyadic Decomposition (CPD) to extract each low-rank 2D matrix, significantly reducing the training cost. `TeZO` can also be easily extended to the `Adam` variant while consuming less memory than `MeZO-SGD`, and requiring about only 35% memory of `MeZO-Adam`. Both comprehensive theoretical analysis and extensive experimental research have validated its efficiency, achieving SOTA-comparable results with lower overhead of training time and memory.

## 1 Introduction

As the model size progresses at an extraordinary rate (Zhang et al., 2022; Touvron et al., 2023; Achiam et al., 2023), memory and computational resources have become the primary bottleneck limiting development. In response to this challenge, ZO has opened up new possibilities for efficient training (Shen et al., 2023). Adopting gradient-free updates with a small amount of high-quality data perfectly unlocks the knowledge of the entire domain, offering significant potential for several practical applications. Since Spall (1992) introduced ZO as a promising alternative to FO in the training process, it has been widely applied in gradient-computation-challenged scenarios (Wang et al., 2018; Liu et al., 2020) and in black-box optimization (Chen et al., 2017; Tu et al., 2019). Recent studies have also highlighted the great potential of ZO in fine-tuning LLMs. Malladi et al. (2023) propose the `MeZO` method which adopts classical `ZO-SGD` (Ghadimi & Lan, 2013) for fine-tuning. Furthermore, it reduces memory costs by only preserving random seeds instead of variables. Compared to FO, it can achieve comparable performance while requiring approximately 10% of memory in practice, greatly improving memory efficiency.

Although ZO has made significant progress, it still faces two challenges, i.e., i) lack of detailed characterization of gradients; ii) the costs of optimization states to generate random variables significantly increase as $d$ grows. This also highlights the bottleneck of ZO methods in LLM tasks. Recent advances learned the strong low-rank nature of gradients in LLMs (Wang et al., 2023; Jaiswal et al., 2024), making low-rank representations in ZO methods as an ingenious solution to the aforementioned issues. With barely compromising performance, low-rank ZO methods effectively reduce the required memory for ZO estimations from $\mathcal{O}(d)$ to $\mathcal{O}(\sqrt{d}r)$ at most, where $r$ is the rankness constant (Chen et al., 2024; Yu et al., 2024). This implementation further endows the ZO method with superior value in the tasks of fine-tuning LLMs.

**Our Motivations.** Existing methods only consider each individual gradient to be low-rank, which cannot naturally extend memory efficiency to other advanced optimizers. In other words, isolated

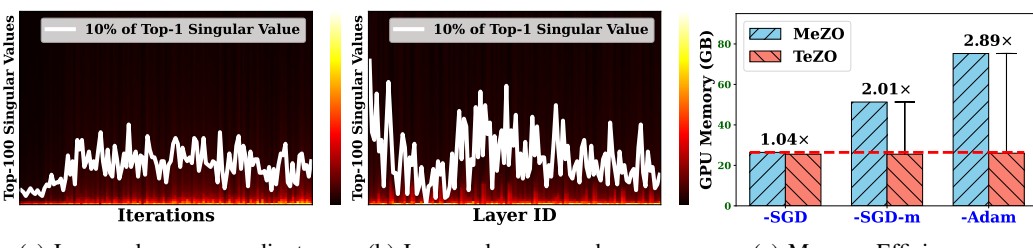

(a) Low-rankness on gradients.  (b) Low-rankness on subspace.  (c) Memory Efficiency.

Figure 1: (a) and (b) are test of the low-rankness of gradients. We finetune OPT-1.3B on SST-2 and calculate top-100 singular values of gradients of `layers.9.self_attn.out_proj.weight`. We then concatenate these singular value vectors and display them as a heat-map in (a). Then we concatenate the normalized gradient of each layer over a total of $T$ iterations into a matrix with the size of $d_l \times T$, calculate the top-100 singular values corresponding to layers and display them as a heat-map in (b). In (c), we record the GPU memory usage of `MeZO`, our `TeZO`, and corresponding variants on training OPT-13B model. We also provide more interesting experiments on the low-rankness and studies of subspace of gradients on LLaMA-7B in Appendix B.1.

gradient low-rankness cannot drive efficient storage and computation of optimizer states. This inspires our contemplation: *how can low-rankness be further incorporated into the ZO optimizer states?*

To investigate an efficient approach to addressing this question, in this paper, we comprehensively study the characteristics of the gradients in LLMs. As shown in *Figure 1.(a) and (b)*, in the tasks of fine-tuning LLMs, the gradients exhibit the following two properties simultaneously: i) the individual gradient at each iteration is approximately low-rank; ii) all gradients along $T$ iterations lie almost within a similar subspace. Obviously, combining properties i) and ii) can lead to higher efficiency. Inspired by this, we propose the `TeZO` estimator to empower the low-rankness on the temporal dimension. Specifically, we estimate the ZO perturbations as a 3D tensor with the size of $m \times n \times T$. By adopting the Canonical Polyadic Decomposition (CPD) (Hitchcock, 1927), the 3D tensor can be estimated by the sum of $r$ rank-1 tensors where $r$ approximates its rank. The joint low-rankness significantly reduces the cost of factor vectors during computation. At each iteration $t$, we only need to generate temporal factor vector to extract a 2D matrix, further lowering the costs from $\mathcal{O}(\sqrt{d}T)$ to $\mathcal{O}(\sqrt{d}+T)$. We also introduce an auxiliary technique to dynamically select the rank $r_l$ for each layer. `TeZO` naturally enables low-rank representations of various optimizer states, thereby facilitating memory-efficient advanced zeroth-order optimizers. As shown in *Figure 1.(c)*, `TeZO-Adam` consumes less memory than `MeZO-SGD`, and requiring about only 35% memory of `MeZO-Adam`. Both comprehensive theoretical analysis and extensive experimental research are conducted to validate its efficiency. `TeZO-Adam` achieves SOTA-comparable performance while requiring only the memory overhead of general `ZO-SGD`. We summarize our contributions as follows:

- By jointly considering low-rankness of gradients and their similarity on LLMs, we propose a novel low-rank ZO estimator, `TeZO`, which constructs the ZO perturbations via CPD to reduce the training overhead, which can be naturally extended to various optimizer states.

- We introduce an auxiliary technique to dynamically select the rank for each layer to further reduce storage requirements. The dynamic rank assigns different rank coefficients to each layer based on the low-rank characteristics of its parameters, providing a fine-grained allocation scheme and effectively avoiding performance degradation caused by improper $r$.

- We prove that `TeZO` is an unbiased estimator of FO gradient, maintaining the comparable variance and convergence rate as existing ZO methods with less memory requirements. Extensive experiments are conducted to validate its efficiency and performance on LLMs.

## 2  RELATED WORK

**Zero-Order Optimization.** Since Spall (1992) proposed the ZO method, it has been extensively studied and practically incorporated in various domains (Chen et al., 2017; Tu et al., 2019; Vemula

et al., 2019; Hajinezhad et al., 2019; Gratton et al., 2021). By avoiding the massive computation and memory requirements of BP, it significantly reduces the training cost while maintaining high performance. As an alternative to FO, it has also been widely explored from several optimization perspectives, e.g. convergence for convex and non-convex (Wang et al., 2018; Golovin et al., 2019; Cheng et al., 2021), non-smooth (Liu et al., 2018a; Kazemi & Wang, 2024; Rando et al., 2024), variance reduction (Liu et al., 2018b; Ji et al., 2019) and primal dual methods (Liu et al., 2018a; Yi et al., 2021; Huang et al., 2024). It has also demonstrated strong potential for applications in certain practical scenarios, e.g. attack and defense (Zhao et al., 2020; Kariyappa et al., 2021), privacy protection (Gratton et al., 2021; Zhang et al., 2023; Gupta et al.), fairness (Chen et al., 2023; Wang et al., 2024b), multi-agent (Tang et al., 2020; Maritan & Schenato, 2023), and efficient training (Nikolakakis et al., 2022; Fang et al., 2022; Mukhoty et al., 2023). These developments highlight the powerful potential of ZO methods in deep learning and artificial intelligence.

**Fine-tuning LLMs with ZO.** In this paper, we focus on the tasks of fine-tuning LLMs. Recent research on LLMs has demonstrated their immense value (Brown et al., 2020; Kojima et al., 2022). However, expensive time and memory costs in the training have become a significant barrier and hinder the research and application (Zhao et al., 2023; Naveed et al., 2023). To unlock the tremendous potential of LLMs, researchers focus more on the training efficiency, leading to significant progress. The application of ZO optimizers has become a shining star from an optimization perspective. Since Malladi et al. (2023) introduce the MeZO method, a series of ZO optimizer variants have been widely developed. Jiang et al. (2024); Yang et al. (2024); Zhao et al. (2024c;a) focus on incorporating adaptivity and curvature information to accelerate ZO optimizers for LLMs. Liu et al. (2024); Guo et al. (2024); Wang et al. (2024a) incorporate the sparsity to further reduce the calculations. Gautam et al. (2024) expand the variance reduction ZO estimator and evaluate its improvements in fine-tuning LLMs. These methods improve ZO methods from the general optimization perspective, yield additional computational and memory overhead. Recently, Yu et al. (2024); Chen et al. (2024) further learn the low-rankness of each single gradient and propose different low-rank ZO estimators. These insightful works have advanced the application of ZO in fine-tuning LLMs.

## 3 PRELIMINARIES

In this section, we introduce notations and review developments of ZO and its recent advances in fine-tuning LLMs. By default and unless stated otherwise, we use lowercase letters to represent 1D vectors, e.g. $z$, uppercase letters to represent 2D matrices, e.g. $Z$, and bold uppercase letters to represent 3D tensors, e.g. $\boldsymbol{Z}$. Scalars are represented as lowercase Greek letters, e.g. $\alpha$. Other special computation symbols will be introduced in detail when they are first mentioned.

**ZO Optimizer.** We consider the general and classical minimization problem:

$$\min_w f(w) \triangleq \mathbb{E}_{\xi \sim \mathcal{D}} \left[ f(w, \xi) \right], \tag{1}$$

where $w \in \mathbb{R}^d$ is the learnable parameters and $\xi$ is the fine-tuning dataset sampled from the distribution $\mathcal{D}$. In this paper, we focus on a classical and widely adopted ZO estimator, *Simultaneous Perturbation Stochastic Approximation* (SPSA) (Spall, 1992). Specifically, SPSA estimates ZO gradient as:

$$\nabla^0 f(w, \xi) = \frac{f(w + \rho z, \xi) - f(w - \rho z, \xi)}{2\rho} z, \tag{2}$$

where $z \sim \mathcal{N}(0, I_d)$ is a random variable and $\rho$ is the perturbation rate. Through two forward passes, it measures the projection component of the true gradient in the direction of the random variable $z$.

**Fine-tuning LLMs with ZO.** MeZO (Malladi et al., 2023) explores the tremendous potential of ZO methods in fine-tuning LLMs. Moreover, to reduce memory usage, it leverages PyTorch's permutation feature in random libs, replacing the storage of all random variables by recording the initial random seed for each iteration, namely the *resampling technique*. This implementation enables the ZO method to achieve up to a $12\times$ memory saving in fine-tuning LLMs. The simple ZO-SGD method is sufficient to achieve performance comparable to FO methods in most tasks.

**Low-rank ZO.** Generally, the parameter dimension of LLMs is extremely large, which constitutes a new bottleneck for the further development of MeZO: training costs of the ZO gradients increase linearly with the model dimension $d$. Furthermore, an important fact in fine-tuning LLMs is also

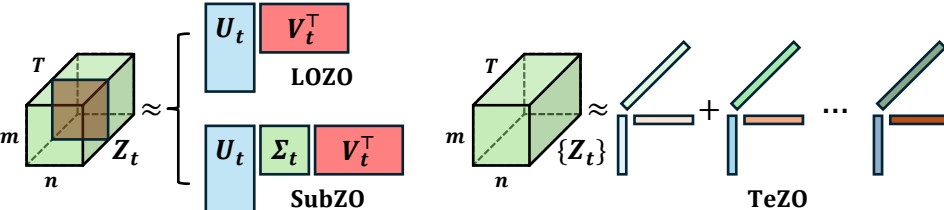

Figure 2: The ZO diagrams for `LOZO`, `SubZO`, and our `TeZO` method. `LOZO` and `SubZO` focus on estimating a single perturbation $Z_t$ as the product of low-rank matrices. `TeZO` construct the entire perturbation set $\boldsymbol{Z} = \{Z_t\}$ via the CPD in the 3D tensor.

ignored: the low-rankness of the gradients. Therefore, the applications of low-rank ZO techniques have emerged. Chen et al. (2024) propose to apply matrix factorization as $Z = UV^\top$ ($Z \in \mathbb{R}^{m \times n}, U \in \mathbb{R}^{m \times r}, V \in \mathbb{R}^{n \times r}$). Additionally, Yu et al. (2024) adopt the form $Z = U\Sigma V^\top$ where $\Sigma \in \mathbb{R}^{r \times r}$, as shown in *Figure 2*. These techniques estimate the low-rank form of each individual perturbation per iteration, reducing the training cost of the ZO method in fine-tuning LLMs. Inspired by these insightful works, we examine another important aspect that is overlooked in the designs of previous works, i.e., *low-rankness on the temporal dimension*. Through the joint low-rank estimation, we propose the `TeZO` method which can further improve the efficiency of the ZO method in fine-tuning LLMs. Further discussions are provided in the next section.

## 4 METHODOLOGY

In this section, we introduce our proposed `TeZO` method. Then we introduce the adaptive selection of the rank of layer-wise gradients. Finally, we demonstrate how `TeZO`, as a structured ZO gradient representation, enables memory-efficient updates of optimizer states.

### 4.1 CANONICAL POLYADIC DECOMPOSITION AND TEZO

*Canonical Polyadic Decomposition* (Hitchcock, 1927), also known as *Parallel Factor Analysis*, is the tensor decomposition technique widely used in data analysis, signal processing, and machine learning. It is a generalization of matrix factorization to higher-order tensors (multi-dimensional arrays). CPD aims to decompose a 3D tensor $\boldsymbol{Z} \in \mathbb{R}^{m \times n \times T}$ into a sum of rank-one tensors and each rank-one tensor is expressed as the outer product of vectors:

$$\boldsymbol{Z} \approx \sum_{s=1}^{r} \chi_s \circ u_s \circ v_s, \tag{3}$$

where $\chi_s \in \mathbb{R}^T, u_s \in \mathbb{R}^m, v_s \in \mathbb{R}^n$ are three factor vectors and $\circ$ denotes the outer product. Based on understanding the low-rank nature along the temporal dimension, we propose a novel low-rank estimation approach to represent the gradient perturbation variables at each iteration, as outlined in *Eq.(3)*. In LLMs, the proportion of 2D model parameters is much larger than that of 1D parameters, so we primarily consider the 2D cases. Specifically, in addition to the conventional factor vectors $u$ and $v$ for the model dimensions, we introduce the factor vectors $\chi$ for the temporal dimension. These three dimensions are independent of each other. Both $u$ and $v$ can be initialized at the beginning of training. Therefore, in the $t$-th iteration, we only need information related to the variable $\tau_s$ without

Table 1: Comparison of `MeZO`, `SubZO`, `LOZO` and `TeZO` on sampling and storage on $W \in \mathbb{R}^{m \times n}$.

| Method | Number of Total Sampling | Number of Storage at Each Iteration |
|---|---|---|
| `MeZO` | $mnT$ | $mn$ |
| `SubZO` | $(m + n + r)rT$ | $mr + nr$ |
| `LOZO` | $(m + n)rT$ | $mr\ /\ nr$ |
| `TeZO` | $(m + n + T)r$ | $r$ |

any additional redundant variables, which can still significantly reduce training costs. Compared to existing studies, we summarize the results in Table 1.

## 4.2 LAYER-WISE SELECTION OF THE RANK $r$

The selection of rank $r$ remains an open challenge. Since ZO methods are typically employed in scenarios where FO gradients are unavailable, it is difficult to directly determine the precise rank of gradients. Recent studies have emphasized the feasibility of low-rank structures, and the rank $r$ is empirically treated as a constant hyperparameter. In fact, $r$ essentially represents a trade-off between performance and efficiency, and selecting an appropriate value for $r$ can significantly enhance the balance between these two factors. Although the constant selection can yield reliable performance, our goal is to identify a more refined solution.

To comprehensively study the rank selection, we learn its connection to the parameters. We dynamically select the rank on different layers in the `TeZO` method. We consider the general cascade layer as: $X_l = \sigma_l(A_l)$, $A_l = W_l X_{l-1} + b_l$, where $\sigma_l(\cdot)$ is activation function, $l \leq L$ is the index of each layer, $W_l$ and $b_l$ are the weight and bias of the $l$-th layer. Therefore, we have:

$$\frac{\partial f}{\partial W_l} = \left( \prod_{p=l+1}^{L} W_p \right)^{\top} \partial \Phi(\sigma_L, \sigma_{L-1}, \cdots, \sigma_l), \tag{4}$$

where $\partial \Phi(\cdot)$ are the joint gradients for all activations from the total mini-batch data samples, whose rank is closely related to the similarity of the input data. In this paper, we focus on the impact from model parameters. According to the rank propagation, the rankness of each gradient satisfies:

$$\text{Rank}\left( \frac{\partial f}{\partial W_l} \right) \leq \text{Rank}\left( \prod_{p=l+1}^{L} W_p \right) \leq \min\left( \text{Rank}\left( W_{l+1} \right), \cdots, \text{Rank}\left( W_L \right) \right). \tag{5}$$

Typically, during training, due to the use of weight decay regularization, the model parameters tend to maintain a high degree of low-rankness. Therefore, the gradients also inherit this property, meaning that the low-rankness of the gradients originates from the low-rankness of the model parameters. We adaptively determine the rank of different layers based on the insight from Eq.(5). In LLMs, there is a natural cascade block structure, where each block contains components such as the attention module and the feed-forward network module. We adopt the truncated Eq.(5) to estimate the rankness of each layer within a block. Specifically, we split $L$ into $B$ blocks as $[\{l_0\}, \{l_1\}, \cdots, \{l_B\}]$. The rankness of the gradient of the $l$-th layer will be estimated as follows:

$$r_l = \min\left( \{\text{Rank}\left( W_{\{l_b\}} \right)\}, r_{max} \right), \tag{6}$$

where $l \in \{l_b\}$ and $r_{max}$ is a constant to prevent the rank selection from becoming excessively large. $\text{Rank}(W) = r_W$ is defined as the largest $r_W$ singular values of the matrix $W$. Generally, in our experiments, we uniformly set a specific threshold to determine $r_W$ that those singular values are larger than the threshold, e.g. approximately 25% of the top-1 singular value. Due to the page limitation, we show the ablation studies in Appendix B.1.4.

## 4.3 TEZO AND ITS EXTENSIONS ON MEMORY EFFICIENT OPTIMIZATION STATES

**TeZO**. In its implementation, we adopt the resampling technique proposed by `MeZO` to reduce memory usage. Before each iteration, the random seed is reset to ensure sampling the same variables. Through three perturbations, we can calculate the positive and negative terms, i.e., $f_+ = f(w + \rho z, \xi)$ and $f_- = f(w - \rho z, \xi)$, and update the projected coefficient $\kappa = (f_+ - f_-)/2\rho$. At each iteration $t$, we only need to sample the component of the temporal factor vector $\tau \in \mathbb{R}^{r_l}$. Then, it updates via the perturbation $G_t = \kappa_t Z_t$ where $Z_t$ can be calculated according to the $t$-th dimension of Eq.(3).

**Memory-Efficient First-order Momentum**. Momentum-based methods typically offer greater stability. However, its drawback lies in the requirement of doubling the storage to maintain the first-order momentum variables in the optimizer state. In contrast, our proposed `TeZO` zeroth-order representation can avoid this issue. When focusing on the first-order momentum of the $l$-th layer:

$$M_t = (1 - \beta_1) \sum_{k=0}^{t} \sum_{s=1}^{r_l} \beta_1^{k-t} \kappa_k (\tau_k)_s (u_s \circ v_s) = \sum_{s=1}^{r_l} \left( (1 - \beta_1) \sum_{k=0}^{t} \beta_1^{k-t} \kappa_k \tau_k \right)_s (u_s \circ v_s). \tag{7}$$

**Algorithm 1** Pipeline of ZO

1: Initialize the rank list $[r_1, \cdots, r_L]$ via Eq.(6)
2: Initialize the factor vectors $\{u_s\}$ and $\{v_s\}$
3: **for** $t = 0, 1, 2, \cdots, T - 1$ **do**
4:     sample the minibatch $\xi_t$ and seed $\zeta_t$
5:     $W = $ **Perturbation**$(W, \rho, \zeta_t, [r_l])$
6:     $f_+ = f(W, \xi)$
7:     $W = $ **Perturbation**$(W, -2\rho, \zeta_t, [r_l])$
8:     $f_- = f(W, \xi)$
9:     $W = $ **Perturbation**$(W, \rho, \zeta_t, [r_l])$
10:     $\kappa_t = (f_+ - f_-)/2\rho$
11:     $W = $ **Update**$(\kappa_t, \zeta_t, \{u_s\}, \{v_s\}, [r_l])$
12: **end for**
13:
14: **Func Perturbation**$(W, \rho, \zeta, [r_l])$:
15: reset the random seed as $\zeta$
16: **for** $W_l \in W$ **do**
17:     sample $\tau \sim \mathcal{N}(0, I_{r_l})$
18:     $Z_t = \sum_{s=1}^{r_l} \tau_s (u_s \circ v_s)$
19:     $W_l = W_l + \rho Z_t$
20: **end for**

**Algorithm 2** Update of TeZO-SGD/M/ADAM

1: **Func Update**$(\kappa, \zeta, \{u_s\}, \{v_s\}, [r_l])$:
2: reset the random seed as $\zeta$
3: **for** $W_l \in W$ **do**
4:     sample factor vector $\tau \sim \mathcal{N}(0, I_{r_l})$
5:     (TeZO-SGD)
6:     $G_t = \sum_{s=1}^{r_l} \kappa \tau_s (u_s \circ v_s)$
7:
8:     (TeZO-M)
9:     $\tau_M = \beta_1 \tau_M + (1 - \beta_1)\kappa\tau$
10:     $G_t = \sum_{s=1}^{r_l} (\tau_M)_s (u_s \circ v_s)$
11:
12:     (TeZO-ADAM)
13:     $\tau_M = \beta_1 \tau_M + (1 - \beta_1)\kappa\tau$
14:     $\tau_V = \beta_2 \tau_V + (1 - \beta_2)\kappa^2\tau^2$
15:     $M_t = \sum_{s=1}^{r_l} (\tau_M)_s (u_s \circ v_s)$
16:     $V_t = \sum_{s=1}^{r_l} (\tau_V)_s (u_s^2 \circ v_s^2)$
17:     $G_t = M_t / \sqrt{V_t + \epsilon}$
18:
19:     $W_l = W_l - \eta_l G_t$
20: **end for**

According to the above equation, the computation of the first-order momentum term can be reformulated by exchanging the order of summation—first performing momentum accumulation on the factor vector $\tau$, and then computing the momentum term via the outer product. As a result, the storage requirement for the optimizer state variable is reduced to $\mathcal{O}(r)$.

**Memory-Efficient Second-order Momentum**. Adaptive optimizer is the mainstream optimization for training LLMs, but its expensive storage requirements impose significant memory pressure. Especially in zeroth-order optimizers, where activations are not stored for gradient computation, the optimizer states become the dominant memory consumption. In contrast, our proposed TeZO method can easily overcome this challenge. When focusing on the second-order momentum of the $l$-th layer:

$$\left(\nabla^0 f(W_l)\right)^2 = \kappa_t^2 \left(\sum_{s=1}^{r_l} \tau_s(u_s \circ v_s)\right)^2 = \underbrace{\sum_{s=1}^{r_l} \kappa_t^2 \tau_s^2(u_s^2 \circ v_s^2)}_{\text{Separable Term}} + \underbrace{\kappa_t^2 \sum_{p \neq q}^{r_l} \tau_p \tau_q (u_p u_q \circ v_p v_q)}_{\approx \mathbf{0}}. \quad (8)$$

The second term, i.e., the cross term, has an overall zero expectation on each coordinate. In practice, this term is approximately zero and negligible. Due to page limitation, we provide more experiments in Appendix B.2. Therefore, TeZO enables nearly lossless second-order momentum computation solely through the update of the separable term. Its accumulation can be viewed as a first-order momentum form applied to the squared factor vectors, and can therefore be computed in a memory-efficient manner using Eq.(7), which reduce the memory overhead from $\mathcal{O}(d)$ to $\mathcal{O}(r)$.

In fact, advanced optimizers that applies first-order and second-order moments can be efficiently computed through the TeZO structure. In the main text, we present comparisons across three commonly used optimizers, and in Appendix B.1.5, we provide further experiments on TeZO-LION.

## 5 THEORETICAL ANALYSIS

In this section, we mainly introduce the theoretical analysis of TeZO, including fundamental properties, convergence guarantees and the memory comparisons in the application of various optimizers.

**Theorem 1 (Expectation and Variance)** *We consider the 2D parameters $W \in \mathbb{R}^{m \times n}$. Its FO gradient is denoted as $\nabla f$ and ZO gradient is denoted as $\nabla^0 f$. When using the TeZO method to*

Table 2: Theoretical memory usage for the computation of the corresponding variables in `Adam`.

| Method | Weights | Gradients | First-order Mome. | Second-order Mome. |
|---|---|---|---|---|
| MeZO | $mn$ | $mn$ | $mn$ | $mn$ |
| MeZO+LoRA | $mn + (m+n)r$ | $(m+n)r$ | $(m+n)r$ | $(m+n)r$ |
| SubZO | $mn$ | $(m+n+r)r$ | $mn$ | $mn$ |
| LOZO | $mn$ | $(m+n)r$ | $mr$ | $mn$ |
| TeZO | $mn$ | $(m+n+1)r$ | $r$ | $r$ |

*estimate the ZO gradient with rank $r$ and a sufficiently small perturbation rate $\rho$, the following holds:*

$$\mathbb{E}_{\tau,u,v}\left[\frac{1}{r}\lim_{\rho\to 0}\nabla^0 f\right] = \nabla f, \quad \mathbb{E}_{\tau,u,v}\|\frac{1}{r}\lim_{\rho\to 0}\nabla^0 f - \nabla f\|^2 = \delta\|\nabla f\|^2, \tag{9}$$

*where $\delta = 1 + mn + \frac{2mn}{r} + \frac{6(m+n)}{r} + \frac{10}{r}$.*

**Remark 1.1** `TeZO` *is an unbiased zero-order estimator and its variance is linearly correlated with the norm of the FO gradient. Moreover, we provide detailed relationships between the variance coefficient $\delta_l$ and the matrix sizes $m_l, n_l$ as well as rank $r_l$. Previous work ([Yu et al., 2024](#)) focuses on the impact of low-rankness on variance from the perspective of the subspace for the quadratic objective. We provide the formal expression under the low-rank representation for a general smooth objective. The variance for low-rank representation is slightly larger than that of the `MeZO` method, i.e. $mn$, remaining within the same order. This indicates that `TeZO` has comparable ability to `MeZO` in practice while requiring significantly less training costs.*

Then we consider the convergence. In this paper, we consider the general non-convex objective with:

**Assumption 1** *$f(\cdot)$ is a smooth objective, i.e., for $\forall x, y \in \mathbb{R}^d$, $\|\nabla f(x,\xi) - \nabla f(y,\xi)\| \le \lambda\|x-y\|$.*

**Assumption 2** *The stochastic gradient is an unbiased estimator with bounded variance, i.e., for each data sample $\xi$, $\mathbb{E}_\xi\left[\nabla f(x,\xi)\right] = \nabla f(x)$, $\mathbb{E}_\xi\|\nabla f(x,\xi) - \nabla f(x)\|^2 \le \sigma^2$.*

These are two commonly adopted assumptions in ZO optimization. Prior works ([Chen et al., 2024](#); [Yu et al., 2024](#)) consistently impose the requirement that some or all factor vectors exhibit column orthogonality. In contrast, our proof does not rely on the need for such additional constraints.

**Theorem 2 (Convergence)** *Without loss of generality, we consider the 2D parameters $W \in \mathbb{R}^{m\times n}$. Under Assumption 1 and 2, let $\eta = \mathcal{O}\left(\sqrt{\frac{D_0}{\lambda T(\rho^2\lambda^2\delta_\rho + \delta\sigma^2)}}\right) \le \frac{1}{\lambda(\delta+1)}$ where $D_0 = f(W_0) - f(W_\star)$ is the initialized bias, the sequence $\{W_t\}_{t=0}^{T-1}$ generated by `TeZO` converges as:*

$$\frac{1}{T}\sum_{t=0}^{T-1}\mathbb{E}\|\nabla f(W_t)\|^2 = \mathcal{O}\left(\sqrt{\frac{\lambda D_0\left(\rho^2\lambda^2\delta_\rho + \delta\sigma^2\right)}{T}}\right), \tag{10}$$

*where $\delta_\rho = \frac{15r^2(m+3)^3(n+3)^3 + 36r^3m^3n^3 + r^4m^3n^3}{4}$ and $\delta$ is defined in Theorem 1.*

**Remark 2.1** *This convergence result maintains the same rate of recent ZO advances. By substituting the total parameters for $d$, we have $\delta = \mathcal{O}(d)$ and $\delta_\rho = \mathcal{O}(d^3)$. Let the perturbation rate $\rho = \mathcal{O}(\frac{\sigma}{\lambda}d^{-1})$, we have the final rate as $\mathcal{O}(\sqrt{\frac{\lambda D_0 d\sigma^2}{T}})$ which recovers the general rate of the recent ZO methods. This also demonstrates the advantages of the `TeZO` method, as it reduces the complexity of random sample generation from $\mathcal{O}(d \cdot T)$ to $\mathcal{O}(\sqrt{d} + T)$ and effectively decreases memory usage, while theoretically maintaining the similar convergence rate.*

**Theoretical Memory Overhead.** We compare the memory requirements to demonstrate the advantage of the `TeZO` in terms of memory efficiency. We mainly focus on the theoretical memory of `MeZO`, `MeZO+LoRA`, `LOZO`, `SubZO` with Adam. By considering a 2D weight $W \in \mathbb{R}^{m\times n}(m \le n)$, we show the results in Table 2. Previous methods solely adopt low-rankness from the perspective of gradients, and thus fail to effectively extend memory efficiency to optimizer states. `TeZO` introduces temporal low-rankness, which can effectively reduces the storage requirements of optimizer states.

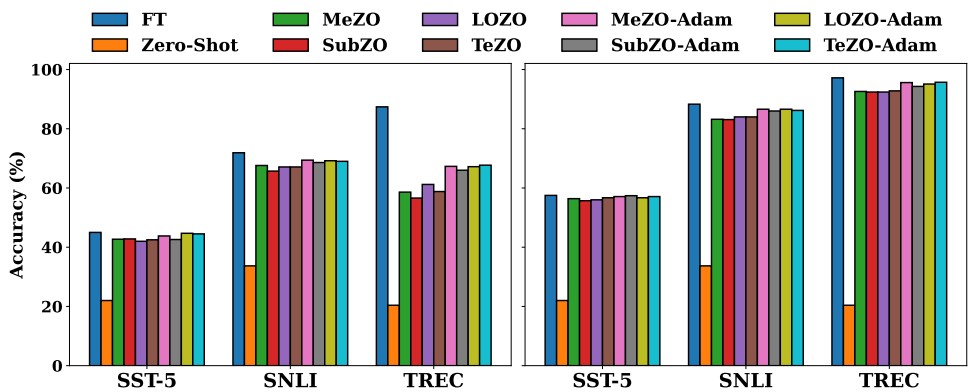

Figure 3: Fine-tuning RoBERTa-large for 80k iterations under $k = 16$ (left) and $k = 512$ (right).

Table 3: Performance and averaged memory usage of fine-tuning OPT-13B for 15k iterations.

| | SST-2 | CB | BoolQ | WIC | M-RC | SQAD | DROP | Memory |
|---|---|---|---|---|---|---|---|---|
| FT | 93.5 | 84.0 | 76.4 | 70.0 | 71.1 | 84.7 | 31.5 | 247.26 G |
| ZERO-SHOT | 58.5 | 46.4 | 59.1 | 55.2 | 46.7 | 46.6 | 14.4 | 26.17 G |
| MEZO | $90.1_{(0.5)}$ | $67.9_{(0.8)}$ | $66.1_{(0.6)}$ | $54.7_{(0.5)}$ | $57.7_{(0.9)}$ | $79.6_{(1.0)}$ | $30.4_{(0.3)}$ | 28.22 G |
| SUBZO | $91.3_{(0.7)}$ | $67.9_{(0.8)}$ | $66.1_{(0.5)}$ | $55.9_{(0.4)}$ | $57.3_{(0.8)}$ | $80.7_{(1.2)}$ | $30.5_{(0.4)}$ | 28.65 G |
| LOZO | $90.3_{(0.6)}$ | $67.9_{(0.8)}$ | $65.6_{(0.5)}$ | $55.3_{(0.5)}$ | $56.9_{(0.9)}$ | $80.5_{(0.9)}$ | $30.1_{(0.8)}$ | 27.39 G |
| TEZO | $90.2_{(0.7)}$ | $69.6_{(0.8)}$ | $65.1_{(0.4)}$ | $54.3_{(0.4)}$ | $56.8_{(0.7)}$ | $80.7_{(0.7)}$ | $29.6_{(0.4)}$ | 27.41 G |
| MEZO-M | $90.6_{(0.5)}$ | $67.9_{(0.8)}$ | $65.5_{(0.5)}$ | $54.6_{(0.3)}$ | $57.9_{(0.9)}$ | $79.5_{(1.1)}$ | $30.4_{(0.6)}$ | 53.07 G |
| SUBZO-M | $91.3_{(0.5)}$ | $67.9_{(0.8)}$ | $65.2_{(0.7)}$ | $54.9_{(0.7)}$ | $57.5_{(0.8)}$ | $80.5_{(1.3)}$ | $30.1_{(0.7)}$ | 55.44 G |
| LOZO-M | $90.7_{(0.4)}$ | $67.1_{(0.6)}$ | $65.7_{(0.8)}$ | $55.7_{(0.5)}$ | $57.7_{(0.8)}$ | $80.7_{(1.1)}$ | $29.9_{(0.5)}$ | 27.44 G |
| TEZO-M | $91.1_{(0.4)}$ | $69.6_{(0.6)}$ | $65.6_{(0.6)}$ | $55.6_{(0.5)}$ | $57.9_{(0.9)}$ | $80.9_{(0.9)}$ | $30.4_{(0.5)}$ | 27.43 G |
| ZO-ADAMU | $92.0_{(0.8)}$ | $67.9_{(0.6)}$ | $71.0_{(0.7)}$ | $59.7_{(0.4)}$ | $59.4_{(0.6)}$ | $82.4_{(0.9)}$ | $\mathbf{31.1}_{(0.6)}$ | 77.12 G |
| MEZO-ADAM | $92.4_{(0.5)}$ | $67.9_{(0.6)}$ | $70.0_{(0.7)}$ | $58.7_{(1.1)}$ | $58.9_{(0.5)}$ | $81.8_{(0.8)}$ | $30.7_{(0.4)}$ | 78.16 G |
| SUBZO-ADAM | $92.8_{(0.5)}$ | $67.9_{(0.8)}$ | $70.3_{(0.6)}$ | $60.3_{(0.5)}$ | $59.9_{(0.5)}$ | $81.3_{(0.8)}$ | $30.5_{(0.3)}$ | 78.85 G |
| LOZO-ADAM | $93.2_{(0.4)}$ | $69.6_{(0.8)}$ | $70.0_{(0.6)}$ | $59.7_{(0.6)}$ | $59.7_{(0.6)}$ | $82.6_{(0.6)}$ | $30.3_{(0.2)}$ | 53.31 G |
| TEZO-ADAM | $\mathbf{93.3}_{(0.5)}$ | $\mathbf{69.6}_{(0.8)}$ | $\mathbf{71.8}_{(0.8)}$ | $\mathbf{60.5}_{(0.7)}$ | $\mathbf{60.3}_{(0.5)}$ | $\mathbf{84.0}_{(1.1)}$ | $30.8_{(0.3)}$ | $\mathbf{28.04}$ G |

# 6 EXPERIMENTS

In this section, we mainly show the empirical studies. We follow the recent studies of fine-tuning LLMs tasks with ZO methods (Malladi et al., 2023; Yu et al., 2024; Chen et al., 2024; Jiang et al., 2024) and adopt the similar setups to validate the efficiency. The main text primarily introduces baselines, performance evaluations, and training costs. Due to page limitations, other contents, including experimental details, hyperparameter selections, have been stated in Appendix B.

**Baselines and setups.** We select recent advances of ZO and low-rank ZO methods on fine-tuning LLM tasks as baselines, including MeZO (Malladi et al., 2023), LOZO (Chen et al., 2024), SubZO (Yu et al., 2024), and their variants of momentum-based and Adam-based extensions in their works. We also compare ZO-AdaMU (Jiang et al., 2024) which focuses on adaptivity. Similar to these works, we conducted tests on different models, including RoBERTa-large (Liu et al., 1907), OPT (Zhang et al., 2022), and LLaMA (Touvron et al., 2023). We select a total of 16 datasets for testing and compute the final average performance to fairly compare the overall efficiency of each method.

**Medium-sized Models.** We conduct the experiments on the RoBERTa-large model for the general sentiment classification, natural language inference and text retrieval tasks, as shown in *Figure 3*. To eliminate experimental randomness, the reported results are the averages of 5 runs with different random seeds. It clearly demonstrates the efficiency of the ZO method on medium-sized models. In fact, for medium-sized models, MeZO remains the most accurate ZO method.

Table 4: Fine-tuning LLaMA-7B and LLaMA-30B for 20k iterations.

| | LLaMA-7B | | | | LLaMA-30B | | | |
| --- | --- | --- | --- | --- | --- | --- | --- | --- |
| | SST-2 | RTE | WIC | AVG. | SST-2 | RTE | WIC | AVG. |
| FT | 95.6 | 86.3 | 70.4 | +0 | 95.9 | 88.2 | 71.1 | +0 |
| ZERO-SHOT | 59.7 | 49.8 | 50.6 | -30.7 | 63.5 | 55.9 | 58.4 | -25.8 |
| MeZO | $93.7_{(0.7)}$ | $69.0_{(0.8)}$ | $60.5_{(0.4)}$ | -9.7 | $93.5_{(0.6)}$ | $69.7_{(0.3)}$ | $63.5_{(0.3)}$ | -9.5 |
| SubZO | $93.1_{(0.8)}$ | $67.9_{(0.8)}$ | $59.3_{(0.5)}$ | -10.6 | $92.8_{(0.6)}$ | $68.4_{(0.4)}$ | $63.0_{(0.4)}$ | -10.3 |
| LOZO | $93.6_{(0.4)}$ | $69.5_{(0.7)}$ | $60.2_{(0.4)}$ | -9.7 | $93.8_{(0.5)}$ | $69.6_{(0.2)}$ | $63.2_{(0.3)}$ | -9.4 |
| TeZO | $92.9_{(0.6)}$ | $67.0_{(0.9)}$ | $59.9_{(0.6)}$ | -10.8 | $94.0_{(0.5)}$ | $69.5_{(0.3)}$ | $64.2_{(0.6)}$ | -9.2 |
| MeZO-Adam | $94.4_{(0.5)}$ | $71.4_{(0.9)}$ | $\mathbf{61.9}_{(0.5)}$ | -8.2 | $94.8_{(0.9)}$ | $72.2_{(0.7)}$ | $64.1_{(0.3)}$ | -8.0 |
| SubZO-Adam | $93.8_{(0.7)}$ | $72.4_{(1.1)}$ | $60.5_{(0.4)}$ | -8.5 | $94.0_{(0.4)}$ | $74.1_{(0.6)}$ | $63.0_{(0.3)}$ | -8.0 |
| LOZO-Adam | $94.6_{(0.5)}$ | $73.4_{(0.8)}$ | $60.6_{(0.7)}$ | -7.9 | $\mathbf{94.8}_{(0.5)}$ | $74.6_{(0.8)}$ | $63.9_{(0.3)}$ | -7.3 |
| TeZO-Adam | $\mathbf{94.6}_{(0.4)}$ | $\mathbf{75.0}_{(1.2)}$ | $60.8_{(0.3)}$ | **-7.3** | $94.7_{(0.3)}$ | $\mathbf{76.5}_{(0.9)}$ | $\mathbf{64.3}_{(0.5)}$ | **-6.5** |

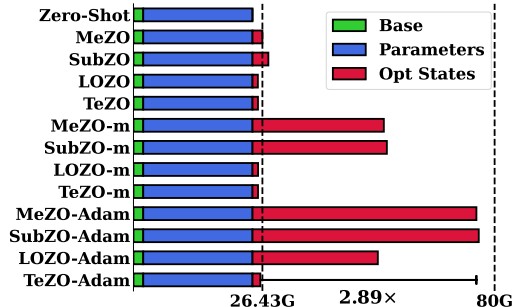 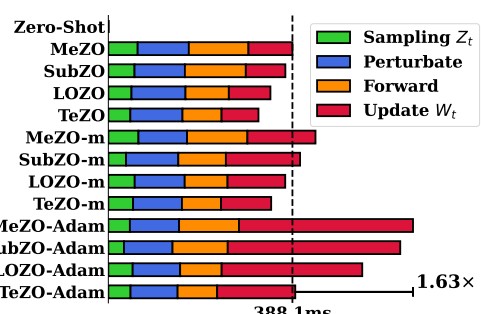

(a) Memory breakdown on fine-tuning OPT-13B.   (b) Wall-time breakdown on fine-tuning OPT-13B.

Figure 4: GPU memory usage (a) and wall-clock time (b) for fine-tuning OPT-13B on the SST-2 dataset on a single H100 device. More experiments are stated in Appendix B.4.

**Large-sized Models.** We conduct experiments on OPT-13B and LLaMA-7B, as shown in *Table 3 4*. The reported results are the averages of 3 runs with different random seeds. The low-rank ZO methods and their variants generally perform better than the vanilla MeZO method. MeZO-m and MeZO-Adam can achieve about 0.2% and 2.1% improvements. Due to the strong low-rank nature of TeZO, the alignment of factor vectors used in adaptivity still retains strong subspace properties. In practical training, the benefit of this advantage is that it constantly enforces the adaptive learning rate to stay synchronized within the subspace. Therefore, TeZO-Adam can still achieve the SOTA-comparable performance, about 2.2% improvement on LLaMA-7B and 2.8% improvement on OPT-13B.

**Memory Usage and Wall-clock Time.** We evaluate the GPU memory usage and wall-clock time for different methods. *Figure 4.* (a) shows the memory cost of ZO mainly consists of two parts, parameters and optimizer states. For the MeZO baseline, -Adam variant typically consumes 3× the storage. However, our proposed TeZO-Adam method requires less storage than MeZO, and is significantly lower than MeZO-Adam (∼34.6%). *Figure 4.* (b) shows the wall-clock time comparisons, primarily including sampling, perturbations, forward pass, and update parameters. our TeZO-Adam maintains a speed comparable to the MeZO and is 1.63× faster than MeZO-Adam.

## 7 CONCLUSION

Inspired by the similarity in the gradient subspace, in this paper, we combine the low-rank properties in both the model and the temporal dimension and propose a novel low-rank ZO method, named TeZO. Moreover, TeZO can easily implement memory-efficient variants of momentum and Adam, maintaining the same resource consumption as standard ZO-SGD, but with better performance. We prove that TeZO maintains the same convergence rate as previous low-rank ZO methods while requiring fewer training costs. Furthermore, we conduct extensive evaluations of TeZO and its variants in fine-tuning tasks of LLMs, which demonstrates the significant potential of low-rank ZO.

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

## A    DECLARATION

In the preparation of this manuscript, large language models were used solely for spelling and grammar correction. Beyond this purpose, no other large-model-based tools were employed. We include a code demo in the supplementary material and provide detailed tables of hyperparameter choices in the main text to support the reproducibility of this work. Moreover, our work focuses purely on algorithmic research, relying solely on commonly used public datasets and models from the research community, and does not involve any other ethical concerns.

## B    EXPERIMENT MATERIALS

### B.1    LOW RANKNESS IN LLMS

This property has been well studied and validated by several works. Especially in large models, the low-rank nature of parameters, gradients, and the FO optimizer states have triggered a series of studies. The most representative works include LoRA low-rank structure (Hu et al., 2021), GaLore low-rank optimization (Zhao et al., 2024b), and so on. In the main text, we study the low-rankness on the OPT-1.3B model. Here, we also show tests of the low-rankness on the LLaMA-7B.

#### B.1.1    LOW RANKNESS OF EACH SINGLE GRADIENT

We first learn the low rankness of each single gradient. Similarly, we consider the 2D parameters $W_l \in \mathbb{R}^{m \times n}$. Then we calculate the top-100 singular values of its gradients $\nabla_{W_l} f \in \mathbb{R}^{m \times n}$ to test the low-rankness, As shown in *Figure 5*.

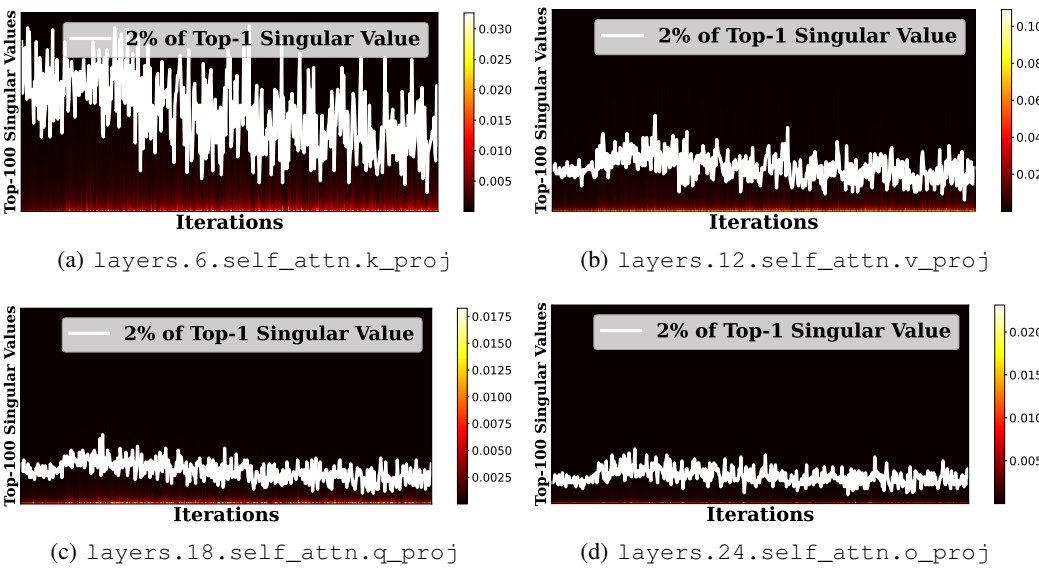

(a) layers.6.self_attn.k_proj

(b) layers.12.self_attn.v_proj

(c) layers.18.self_attn.q_proj

(d) layers.24.self_attn.o_proj

Figure 5: We finetune LLaMA-7B on SST-2 to test low-rankness of gradients. We set the batchsize as 16 and train 500 steps with 8000 samples on a H200 device. The training loss decreases from 1.04 to 0.13. We analyze the low-rank properties of the $W_K$, $W_V$, $W_Q$ and $W_O$ parameters in the 6-th, 12-th, 18-th, and 24-th modules at each iteration ($W_K, W_V, W_Q, W_O \in \mathbb{R}^{4096 \times 4096}$). The white lines represent the indices where the singular values are 2% of the maximum singular value.

It is clear that gradients are low-rank on LLaMA-7B model and the low-rankness is even greater than that of OPT-1.3B. After around index-20, the singular value is almost completely lost. It is worth noting that in our tests, **the data samples used for each gradient computation are completely different**, which further emphasizes the universality of gradient low-rankness in LLMs.

### B.1.2 LOW RANKNESS OF GRADIENT SUBSPACE

The low-rankness of each individual gradient has already been widely acknowledged. In this part, we continue to explore the low-rank subspace of all gradients in the LLaMA-7B model. The same, we consider the 2D parameters $W_l \in \mathbb{R}^{m \times n}$ trained for $T$ iterations. We normalize each flattened gradient and concatenate them along the $T$ dimension to form a new matrix as $G = [g_{w_l,0}, g_{w_l,1}, \cdots, g_{w_l,T}] \in \mathbb{R}^{mn \times T}$ where $g_{w_l,t} = \nabla_{w_l} f_t / \|\nabla_{w_l} f_t\| \in \mathbb{R}^{mn}$. And then we calculate the cosine value by $G^\top G$. It is important to note that **without normalization, the low-rankness of this matrix naturally holds.** This is because when the loss is large, the gradients are naturally large. As training progresses and the loss becomes smaller, the gradients will be much smaller. If these gradients are concatenated directly, although it still forms a low-rank matrix, this low-rank nature is inconsistent with the motivation behind our proposed `TeZO` method. `TeZO` expects similarity across the entire gradient space. Since we are always more concerned with whether the gradient direction is similar, we study the properties of each normalized gradient, specifically whether all gradients lie in the same subspace, as shown in *Figure 6*.

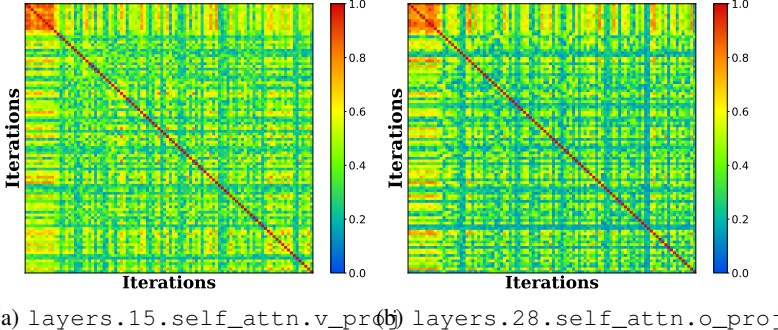

(a) `layers.15.self_attn.v_proj`  (b) `layers.28.self_attn.o_proj`

Figure 6: We finetune LLaMA-7B on SST-2 to test the similarity between gradients at different iterations. Similarly, we set the batchsize as 16 and train 500 steps with 8000 samples on a H200 device. The training loss decreases from 1.04 to 0.13. We calculate the cosine value of each gradient pair $(\nabla_{W_l} f_{t_1}, \nabla_{W_l} f_{t_2})$ where $t_1, t_2 \in [0, 1, 2, \cdots, 499]$ and show their values as the heat maps above.

It can be seen that the similarity between gradients is generally high, and the distribution of cosine distances is relatively concentrated. This also highlights the low-rankness of the gradient space in training LLMs, where gradients from different samples exhibit strong similarity.

### B.1.3 LOW-RANKNESS BETWEEN WEIGHTS AND GRADIENTS

In this part, we explore the close relationship between the low-rankness of gradients and that of model parameters.

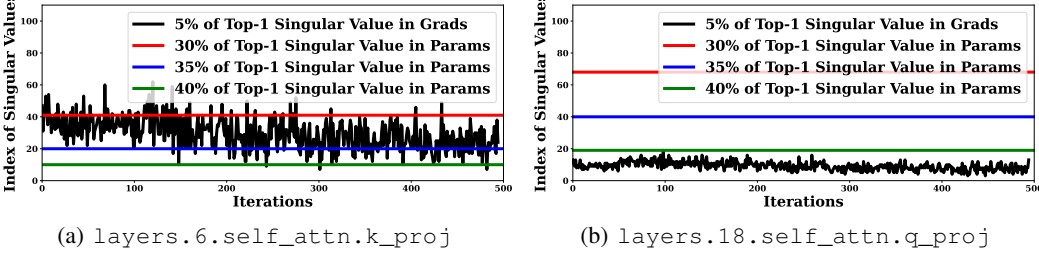

(a) `layers.6.self_attn.k_proj`  (b) `layers.18.self_attn.q_proj`

Figure 7: We finetune LLaMA-7B on SST-2 to test the similarity between gradients at different iterations. We compared the relationship between the low-rank properties of parameters and gradients, and demonstrat the rank levels of parameters.

We can observe that, although the degree of low-rankness in gradients and parameters is not strictly aligned, they still exhibit a high degree of correlation within a certain fluctuation range. In fact, due to the very small learning rate, the gradient updates for individual parameters are negligible, which allows the low-rank nature of the model to remain relatively stable. This also validates the effectiveness of our dynamic selection method within a certain range. The dynamic selection method eliminates the need for additional hyperparameter tuning while ensuring experimental stability.

### B.1.4  COMPARISON BETWEEN DYNAMIC RANKING AND STATIC RANKING

In this section, we mainly compare the differences between dynamic ranking and static ranking. By examining these two approaches, we aim to highlight their respective strengths and limitations, and emphasize that dynamic ranking offers superior adaptability and personalization. We conducted ablation studies on TeZO-Adam for training OPT-13B, 15K iterations.

Table 5: Comparison between dynamic ranking and static ranking.

|  | SST2 | MultiRC | BoolQ | WIC | Averaged $r$ |
|---|---|---|---|---|---|
| $r = 8$ | 89.7 (0.3) | 57.1 (0.2) | 69.4 (0.6) | 59.2 (0.5) | 8 |
| $r = 256$ | 92.9 (0.6) | 59.7 (0.3) | 71.3 (0.6) | 60.1 (0.6) | 256 |
| $r = 1024$ | 92.8 (0.5) | 59.5 (0.7) | 71.2 (0.7) | 60.3 (0.5) | 1024 |
| dynamic | 93.3 (0.5) | 60.3 (0.5) | 71.8 (0.8) | 60.9 (0.7) | 31.52 |

As shown in Table 5, the results above present our evaluations under various fixed ranks, where we observe that relatively large fixed ranks are often required to achieve competitive performance. Although the overall trend suggests the existence of an optimal rank choice, such a selection is difficult to determine in practice and may vary considerably across different datasets. As discussed in the paper, different layers and parameters demonstrate varying degrees of low-rank characteristics. To better capture this layer-wise heterogeneity, we propose a dynamic, layer-wise rank selection strategy. This adaptive mechanism aligns rank allocation more closely with the model's intrinsic structure, leading to improved efficiency and enhanced performance.

More importantly, compared with static settings, dynamic rank selection can substantially reduce computational complexity. This is because most layers do not require a large rank, and only a subset of critical layers need higher ranks to maintain strong performance.

### B.1.5  APPLICABILITY TO OTHER ADVANCED OPTIMIZERS

Advanced optimizers typically rely on low-rank accumulation of gradients or their squared values. For example, the recently proposed LION optimizer applies the sign function after first-order momentum updates. This structure aligns well with our theoretical formulation, and `TeZO` can be directly extended to such optimizers by applying the same low-rank estimation principles. We evaluated the fine-tuning performance of the `ZO-LION` optimizer corresponding to the baseline.

Table 6: Applicability to `ZO-LION` optimizer.

|  | SST2 | RTE | WIC |
|---|---|---|---|
| `MeZO-LION` | 92.1 (0.2) | 72.4 (0.4) | 59.2 (0.6) |
| `SubZO-LION` | 90.2 (0.3) | 72.7 (0.5) | 60.3 (0.2) |
| `LOZO-LION` | 92.1 (0.2) | 73.1 (0.6) | 59.9 (0.3) |
| `TeZO-LION` | 92.5 (0.2) | 73.4 (0.4) | 60.5 (0.5) |

As shown in Table 6, `Tezo` exhibits consistently stable performance under low-rank settings, benefiting from substantially reduced memory consumption. Furthermore, its dynamic rank adaptation effectively preserves model accuracy across diverse training scenarios. we also find `Tezo` delivers strong and reliable improvements when combined with the recent `LION` optimizer. These results indicate that the effectiveness of TeZO is not restricted to a particular optimization algorithm, but generalizes well across a wide range of optimizers, further underscoring its practicality and versatility.

## B.2 THE EFFICIENCY OF LIGHTWEIGHT SECOND-ORDER MOMENTUM IN TeZO-ADAM

In this paper, we propose a lightweight variant to address the storage issue of second-order momentum in the `TeZO-Adam` variant:

$$
\left[\nabla^0 f(w_t)\right]^2 = \kappa_t^2 \left(\sum_{s=1}^{r_l} \tau_s \cdot (u_s \circ v_s)\right)^2 = \underbrace{\sum_{s=1}^{r_l} \kappa_t^2 \tau_s^2 \cdot (u_s^2 \circ v_s^2)}_{\text{Separable Term}} + \underbrace{\kappa_t^2 \sum_{p \neq q}^{r_l} \tau_p \tau_q \cdot (u_p u_q \circ v_p v_q)}_{\mathbb{E}_{\tau,u,v}[\tau_p \tau_q \cdot (u_p u_q \circ v_p v_q)] = \mathbf{0}}.
$$

(11)

The separable term is memory-efficient which can be calculated by the accumulation of the factor vector $\tau$.

### B.2.1 ERRORS IN ONE STEP

By the definitions, we consider the decomposition of $Z \in \mathbb{R}^{m \times n}$ by the factor vectors $u_s \in \mathbb{R}^m$, $v_s \in \mathbb{R}^n$, and $\tau \in \mathbb{R}^r$. And we consider an example comparable in scale to the LLaMA-7B model and set $m = n = 4096$. And we select $r = 64$ to evaluate the error. Since we consider the parameters at time $t$ where $\kappa_t$ can be treated as a constant. Without loss of generality, we set $\kappa_t = 1$ directly and examine the error by randomly sampling $\tau, u, v$, as shown in the following figure.

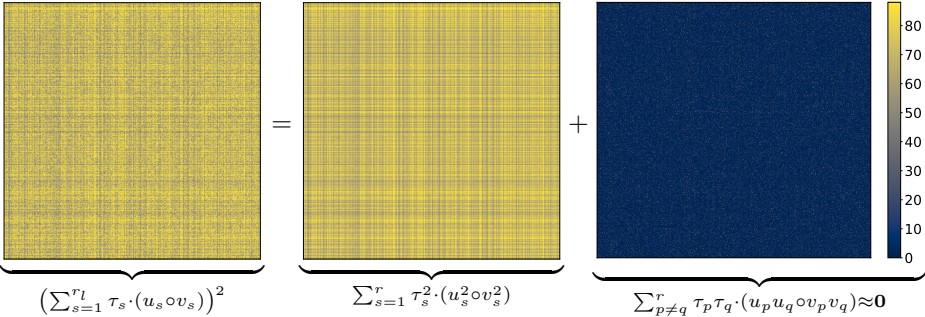

$$\left(\sum_{s=1}^{r_l} \tau_s \cdot (u_s \circ v_s)\right)^2 \qquad \sum_{s=1}^{r} \tau_s^2 \cdot (u_s^2 \circ v_s^2) \qquad \sum_{p \neq q}^{r} \tau_p \tau_q \cdot (u_p u_q \circ v_p v_q) \approx \mathbf{0}$$

This clearly demonstrates the precision of our lightweight estimation. The second term is almost zero, and the cost of calculating it is very high, including both storage and computation. Therefore, we eliminate the second term directly and accumulate the second-order momentum of the first term as the second-order momentum of `Adam` for updates. This significantly reduces the training cost, making the training overhead of our `TeZO-Adam` method almost consistent with that of `MeZO-SGD`, significantly lower than the `MeZO-Adam` method.

### B.2.2 ACCUMULATED ERRORS AFTER $T$ STEPS

Then we learn the accumulated errors in the training process. We define the update of standard second-order momentum as $V_{t+1} = \beta_2 V_t + (1 - \beta_2)\left(\sum_{s=1}^{r} \tau_{s,t} \cdot (u_s \circ v_s)\right)^2$, and that of `TeZO-Adam` as $\hat{V}_{t+1} = \beta_2 \hat{V}_t + (1 - \beta_2)\sum_{s=1}^{r} \tau_{s,t}^2 \cdot (u_s^2 \circ v_s^2)$. We report the averaged accumulated errors $E_t = (V_t - \hat{V}_t)/mn$ over 1000 steps under $\beta_2 = 0.99$, as shown in *Figure 8*.

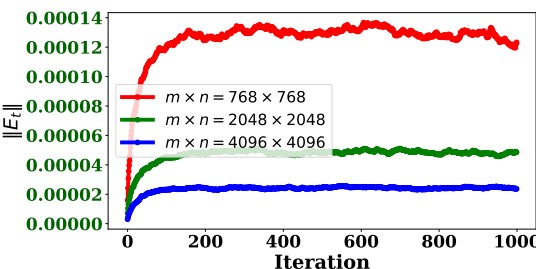

Figure 8: $\|E_t\|$ under different $m, n$ and $r = 64$. It can be observed that the averaged accumulated errors decrease as the model size increases, which highlights the practicality of our proposed lightweight second-order moment estimation on LLMs.

## B.3 SETUPS AND HYPERPARAMETERS

We follow previous works (Malladi et al., 2023; Chen et al., 2024; Yu et al., 2024) and summarized the range of hyperparameter selections, as shown in *Table 7*. Although certain hyperparameters, such as batchsize and perturbation rate, may introduce subtle variations, we fix these hyperparameters across all methods to ensure fairness. The primary search hyperparameter is the learning rate across models of different scales.

Table 7: Hyperparameter recommendations for different models. bs: batchsize; lr: learning rate; $\rho$: perturbation rate; $r$: rank; $f_u$: lazy update interval; $r_{\text{th}}$: threshold for $r$; $r_{\max}$: upper bound of $r$.

| Method | | Search Range | RoBERTa-large | OPT-13B | LLaMA-7B |
|---|---|---|---|---|---|
| MeZO MeZO-m | bs | {16,32,64} | 64 | 16 | |
| | lr | {1e-4, 1e-5, 1e-6, 1e-7} | 1e-6 | 1e-7 | 1e-6 |
| | $\rho$ | 1e-3 | 1e-3 | | |
| MeZO-Adam | bs | {16,32,64} | 64 | 16 | |
| | lr | {1e-4, 3e-5, 1e-5, 3e-6} | - | 1e-5 | 3e-5 / 1e-5 |
| | $\rho$ | 1e-3 | 1e-3 | | |
| SubZO | bs | {16,32,64} | 64 | 16 | |
| | lr | {1e-4, 1e-5, 1e-6, 1e-7} | 1e-6 | 1e-7 | 1e-6 |
| | $\rho$ | 1e-3 | 1e-3 | | |
| | $r$ | {32, 64, 128} | 64 | | |
| | $f_u$ | {50, 100, 500} | 500 | | |
| LOZO LOZO-m | bs | {16,32,64} | 64 | 16 | |
| | lr | {1e-4, 1e-5, 1e-6, 1e-7} | 1e-6 | 1e-7 | 1e-6 |
| | $\rho$ | 1e-3 | 1e-3 | | |
| | $r$ | {8, 16, 32} | 8 | | |
| | $f_u$ | {50, 100, 500} | 100 | | |
| TeZO TeZO-m | bs | {16,32,64} | 64 | 16 | |
| | lr | {1e-4, 1e-5, 1e-6, 1e-7} | 1e-6 | 1e-7 | 1e-6 |
| | $\rho$ | 1e-3 | 1e-3 | | |
| | $r_{\text{th}}$ | {20%, 25%, 30%, 35%} | 25% / 30% | | |
| | $r_{\max}$ | {32, 64, 128, 256} | depend on tasks | | |
| TeZO-Adam | bs | {16,32,64} | 64 | 16 | |
| | lr | {1e-4, 3e-5, 1e-5, 3e-6} | - | 1e-5 | 3e-5 / 1e-5 |
| | $\rho$ | 1e-3 | 1e-3 | | |
| | $r_{\text{th}}$ | {20%, 25%, 30%, 35%} | 25% / 30% | | |
| | $r_{\max}$ | {32, 64, 128, 256} | depend on tasks | | |

We refer to the selections reported in previous works and grid search each hyperparameter. Although further fine-tuning of hyperparameters for specific tasks could yield greater benefits, we fix the hyperparameter selections for fairness. The recommended value reported in the table above is provided only as a reference on which most tasks work well.

## B.4 MEMORY USAGE AND WALL-CLOCK TIME ON DIFFERENT MODEL SIZES

We extensively test the training efficiency of the OPT and LLaMA models across different model sizes, as shown in *Table 8* and *Table 9*.

Table 8: GPU memory usage (max memory reserved) for fine-tuning LLMs on RTE dataset on a single H100 device.

| | OPT | | | | | LLaMA | | |
|---|---|---|---|---|---|---|---|---|
| | 1.3B | 2.7B | 6.7B | 13B | 30B | 7B | 13B | 30B |
| Zero-Shot | 2.90 G | 5.44 G | 12.73 G | 24.39 G | 56.46 G | 12.92 G | 24.89 G | 61.86 G |
| MeZO | 3.48 G | 6.40 G | 14.40 G | 26.43 G | 60.31 G | 13.91 G | 26.12 G | 63.77 G |
| SubZO | 3.28 G | 5.92 G | 14.91 G | 26.97 G | 61.18 G | 14.28 G | 26.67 G | 64.45 G |
| LOZO | 3.31 G | 5.95 G | 13.66 G | 25.50 G | 57.93 G | 13.44 G | 25.77 G | 62.38 G |
| TeZO | 3.28 G | 5.92 G | 13.68 G | 25.52 G | 57.95 G | 13.47 G | 25.79 G | 62.40 G |
| MeZO-m | 6.31 G | 11.77 G | 27.19 G | 51.32 G | >80 G | 26.85 G | 51.31 G | >80 G |
| LOZO-m | 3.32 G | 5.97 G | 13.68 G | 25.53 G | 57.99 G | 13.47 G | 25.80 G | 62.44 G |
| TeZO-m | 3.29 G | 5.93 G | 13.69 G | 25.52 G | 57.96 G | 13.48 G | 25.79 G | 62.40 G |
| MeZO-Adam | 9.15 G | 16.90 G | 39.98 G | 75.27 G | >80 G | 39.50 G | 75.69 G | >80 G |
| TeZO-Adam | 3.48 G | 6.16 G | 14.07 G | 26.01 G | 58.64 G | 13.71 G | 26.16 G | 62.80 G |

Table 9: Wall-clock time per iteration for fine-tuning LLMs on RTE dataset on a single H100 device.

| | OPT | | | | | LLaMA | | |
|---|---|---|---|---|---|---|---|---|
| | 1.3B | 2.7B | 6.7B | 13B | 30B | 7B | 13B | 30B |
| Zero-Shot | - | - | - | - | - | - | - | - |
| MeZO | 69 ms | 111 ms | 212 ms | 388 ms | 871 ms | 212 ms | 372 ms | 942 ms |
| SubZO | 75 ms | 121 ms | 211 ms | 373 ms | 939 ms | 218 ms | 385 ms | 988 ms |
| LOZO | 71 ms | 109 ms | 191 ms | 341 ms | 745 ms | 195 ms | 350 ms | 832 ms |
| TeZO | 67 ms | 106 ms | 178 ms | 316 ms | 680 ms | 186 ms | 325 ms | 775 ms |
| MeZO-m | 76 ms | 123 ms | 236 ms | 437 ms | - | 236 ms | 422 ms | - |
| LOZO-m | 78 ms | 104 ms | 181 ms | 312 ms | 677 ms | 180 ms | 316 ms | 759 ms |
| TeZO-m | 70 ms | 100 ms | 172 ms | 303 ms | 653 ms | 176 ms | 308 ms | 738 ms |
| MeZO-Adam | 104 ms | 173 ms | 348 ms | 642 ms | - | 342 ms | 624 ms | - |
| TeZO-Adam | 92 ms | 134 ms | 224 ms | 394 ms | 841 ms | 227 ms | 397 ms | 937 ms |

From the perspective of memory, low-rank methods have consistently been effective in reducing memory usage. Whether on the OPT or LLaMA models, our TeZO-Adam method consistently incurs lower loss compared to the standard MeZO method, and uses approximately 30% of the memory consumed by the MeZO-Adam method.

From the perspective of wall-clock time, low-rank methods show a significant efficiency improvement on large models, while they perform poorly or even slower on smaller models. On the 125M model, low-rank methods is slower and on the 1.3B models, low-rank methods performs the same as MeZO. Since the model parameters are relatively small, the additional overhead of low-rank computation offsets the training cost. However, when the model size exceeds 3B, the efficiency improvement of low-rank methods becomes significant. Tests on both OPT and LLaMA models show that TeZO-Adam can achieve the same speed as MeZO, while being more than $1.5\times$ faster than MeZO-Adam.

These results are consistent with the *Figure 4* in the main text. From the perspective of computational efficiency, we recommend: it is better to adopt low-rank ZO methods on models larger than 3B to achieve valid improvements.

## B.5 Low-rank Parameters v.s. Low-rank ZO Methods

Gradient low-rank approximation and model low-rank factorization are two key techniques for efficient training, as we mentioned earlier. Techniques like LoRA (Hu et al., 2021) and GaLore (Zhao et al., 2024b), they reduce the number of trainable model parameters and optimizer states through low-rank mapping and subspace mapping, respectively, thereby accelerating the training process. We want to emphasize that these two methods are orthogonal because they target different parameters, addressing the efficient training of different parts of the models during training. Recent works (Yu et al., 2024; Chen et al., 2024) apply low-rank ZO methods to train LoRA models, achieving some success. Here, we would like to emphasize that, according to the experimental records in Appendix B.4, when the size of trainable parameters is too small, low-rank ZO methods provide almost no benefits. For instance, the LoRA model for the 13B model has approximately 300M parameters, and applying low-rank ZO at this parameter scale is clearly unnecessary. Therefore, in this part, we consider these two techniques as independent methods for comparisons, as shown in *Table 10*. The other setups are the same as above.

Table 10: GPU memory usage (max memory reserved) for full fine-tuning, fine-tuning LoRA, fine-tuning prefix, and ZO methods.

|  |  | OPT-6.7B | | OPT-13B | |
|---|---|---|---|---|---|
|  |  | Memory | Ratio | Memory | Ratio |
| FO | ft | 105.24 G | 8.27× | 238.26 G | 9.77× |
|  | ft-LoRA | 37.96 G | 2.98× | 73.19 G | 3.00× |
|  | ft-prefix | 38.23 G | 3.00× | 73.13 G | 3.00× |
| ZO | MeZO | 14.40 G | 1.13× | 26.43 G | 1.08× |
|  | MeZO-LoRA | 13.04 G | 1.02× | 24.82 G | 1.02× |
|  | MeZO-prefix | 13.06 G | 1.03× | 24.81 G | 1.02× |
|  | MeZO-Adam | 39.98 G | 3.14× | 75.27 G | 3.09× |
|  | TeZO-Adam | 14.07 G | 1.10× | 26.01 G | 1.06× |
|  | Zero-Shot | 12.73 G | 1× | 24.39 G | 1× |

Compared to FO methods, the advantages of ZO methods remain significant. Even with methods of low-rank parameters, the memory usage is still nearly three times higher than ZO methods. Additionally, we want to emphasize that while "ZO + LoRA" can further reduce training costs, the gains of memory-efficiency are negligible. Moreover, based on the experiments in existing studies, the performance of these approaches will significantly degrade on large models. "ZO + fine-tuning full parameters" has already achieved to the comparable memory usage of zero-shot (inference only), and combining ZO with LoRA can only save very limited memory. Therefore, we do not advocate directly combining ZO methods with PEFT approaches. From the perspective of memory usage, the benefits of such a combination are indeed limited.

Table 11: Performance and efficiency of full parameter v.s. LoRA on LLaMA-7B.

| Model | Optimizer | SST-2 | Memory | WIC | Memory |
|---|---|---|---|---|---|
| Full | FO+Adam | 95.6 | 115.33 G | 70.4 | 127.20 G |
| LoRA | FO+Adam | 93.2 | 38.25 G | 61.0 | 43.18 G |
| Full | MeZO+Adam | 94.4 | 39.50 G | 61.9 | 44.60 G |
| LoRA | MeZO+Adam | 92.9 | 17.15 G | 59.7 | 20.33 G |
| Full | TeZO+Adam | 94.6 | 13.71 G | 60.8 | 15.67 G |

## C   PROOFS OF MAIN THEOREMS.

### C.1   PROOFS OF THEOREM 1

We consider the mean at first. According to Proposition A.1 proposed by Chen et al. (2024), we have:

$$\lim_{\rho \to 0} \frac{f(W + \rho Z, \xi) - f(W, \xi) - \langle \nabla f(W, \xi), \rho Z \rangle}{\rho} = 0.$$

Without loss of generality, we consider the case where the parameters are 2D matrix. On each step, we sample $\tau \sim \mathcal{N}(0, I_r)$ and compute the perturbation $Z = \sum_{s=1}^{r} \tau_s \cdot (u_s \circ v_s)$. By directly expanding the ZO gradient in $\texttt{TeZO}$, we have:

$$\lim_{\rho \to 0} \nabla^0 f(w, \xi) = \lim_{\rho \to 0} \frac{f(W + \rho Z, \xi) - f(W - \rho Z, \xi)}{2\rho} \cdot Z$$

$$= \lim_{\rho \to 0} \frac{f(W + \rho Z, \xi) - f(W, \xi) - \langle \nabla f(W, \xi), \rho Z \rangle}{2\rho} \cdot Z$$

$$- \lim_{\rho \to 0} \frac{f(W - \rho Z, \xi) - f(W, \xi) - \langle \nabla f(W, \xi), -\rho Z \rangle}{2\rho} \cdot Z$$

$$+ \lim_{\rho \to 0} \frac{\langle \nabla f(W, \xi), \rho Z \rangle}{\rho} \cdot Z = \langle \nabla f(W, \xi), Z \rangle \cdot Z,$$

where the inner product performs as the calculation in vectors. With $Z$ substituted, the following holds:

$$\lim_{\rho \to 0} \nabla^0 f(W, \xi) = \left\langle \nabla f(W, \xi), \sum_{s=1}^{r} \tau_s \cdot (u_s \circ v_s) \right\rangle \cdot \sum_{s=1}^{r} \tau_s \cdot (u_s \circ v_s).$$

Specifically, we consider the element $\left[ \lim_{\rho \to 0} \nabla^0 f(w, \xi) \right]_{i^\star, j^\star}$. To simplify the expression, we have slightly abused the notation $u_{s,i}$ and $v_{s,j}$, which means the $i$-th element in vector $u_s$ and $j$-th element in vector $v_s$. By taking the expectation,

$$\mathbb{E} \left[ \lim_{\rho \to 0} \nabla^0 f(W, \xi) \right]_{i^\star, j^\star} = \mathbb{E} \left\langle \nabla f(W, \xi), \sum_{s=1}^{r} \tau_s \cdot (u_s \circ v_s) \right\rangle \cdot \sum_{s=1}^{r} \tau_s u_{s, i^\star} v_{s, j^\star}$$

$$= \mathbb{E} \sum_{i,j} \left( \nabla f(W, \xi)_{i,j} \sum_{s=1}^{r} \tau_s u_{s,i} v_{s,j} \right) \cdot \sum_{s=1}^{r} \tau_s u_{s, i^\star} v_{s, j^\star}$$

$$= \mathbb{E} \underbrace{\sum_{i \neq i^\star, j \neq j^\star} \nabla f(W, \xi)_{i,j} \sum_{s,s'}^{r} \tau_s \tau_{s'} u_{s,i} u_{s', i^\star} v_{s,j} v_{s', j^\star}}_{\mathbb{E}_{u,v} \left[ u_{s,i} u_{s', i^\star} v_{s,j} v_{s', j^\star} \right] = 0} + \mathbb{E} \underbrace{\sum_{i = i^\star, j \neq j^\star} \nabla f(W, \xi)_{i,j} \sum_{s,s'}^{r} \tau_s \tau_{s'} u_{s,i} u_{s', i} v_{s,j} v_{s', j^\star}}_{\mathbb{E}_v \left[ v_{s,j} v_{s', j^\star} \right] = 0}$$

$$+ \mathbb{E} \underbrace{\sum_{i \neq i^\star, j = j^\star} \nabla f(W, \xi)_{i,j} \sum_{s,s'}^{r} \tau_s \tau_{s'} u_{s,i} u_{s', i^\star} v_{s,j} v_{s', j}}_{\mathbb{E}_u \left[ u_{s,i} u_{s', i^\star} \right] = 0} + \nabla f(W, \xi)_{i^\star, j^\star} \mathbb{E} \sum_{s,s'}^{r} \tau_s \tau_{s'} u_{s, i^\star} u_{s', i^\star} v_{s, j^\star} v_{s', j^\star}$$

$$= \nabla f(W, \xi)_{i^\star, j^\star} \mathbb{E} \underbrace{\sum_{s \neq s'}^{r} \tau_s \tau_{s'} u_{s, i^\star} u_{s', i^\star} v_{s, j^\star} v_{s', j^\star}}_{\mathbb{E}_\tau [\tau_s \tau_{s'}] = 0} + \nabla f(W, \xi)_{i^\star, j^\star} \mathbb{E} \underbrace{\sum_{s=1}^{r} \tau_s^2 u_{s, i^\star}^2 v_{s, j^\star}^2}_{= r} = r \nabla f(W, \xi)_{i^\star, j^\star}.$$

Clearly, when the SPSA form is directly applied, the expectation of the $\texttt{TeZO}$ gradient becomes $r$ times the FO gradient. Therefore, by dividing $r$, $\texttt{TeZO}$ is an unbiased estimation of the FO gradient.

Then we consider the variance. We have the following term:

$$\mathbb{E} \| \frac{1}{r} \lim_{\rho \to 0} \nabla^0 f(w, \xi) - \nabla f(W, \xi) \|^2 = \frac{1}{r^2} \mathbb{E} \| \lim_{\rho \to 0} \nabla^0 f(w, \xi) \|^2 - \mathbb{E} \| \nabla f(W, \xi) \|^2$$

$$= \frac{1}{r^2} \mathbb{E} \| \left\langle \nabla f(W, \xi), \sum_{s=1}^{r} \tau_s \cdot (u_s \circ v_s) \right\rangle \cdot \sum_{s=1}^{r} \tau_s \cdot (u_s \circ v_s) \|^2 - \mathbb{E} \| \nabla f(W, \xi) \|^2$$

$$= \frac{1}{r^2} \mathbb{E} \underbrace{\left\langle \nabla f(W, \xi), \sum_{s=1}^{r} \tau_s \cdot (u_s \circ v_s) \right\rangle^2}_{A} \cdot \underbrace{\left\langle \sum_{s=1}^{r} \tau_s \cdot (u_s \circ v_s), \sum_{s=1}^{r} \tau_s \cdot (u_s \circ v_s) \right\rangle}_{B} - \mathbb{E} \| \nabla f(W, \xi) \|^2.$$

Let $g_{ij} = \nabla f(W, \xi)_{i,j}$ for convenience, we have:

$$A = \left\langle \nabla f(W, \xi), \sum_{s=1}^{r} \tau_s \cdot (u_s \circ v_s) \right\rangle^2 = \sum_{i,i'} \sum_{j,j'} \sum_{s,s'} g_{i,j} g_{i',j'} \tau_s \tau_{s'} u_{s,i} u_{s',i'} v_{s,j} v_{s',j'}$$

$$= \underbrace{\sum_{i \neq i'} \sum_{j \neq j'} \sum_{s \neq s'} g_{i,j} g_{i',j'} \tau_s \tau_{s'} u_{s,i} u_{s',i'} v_{s,j} v_{s',j'}}_{A_1} + \underbrace{\sum_{i \neq i'} \sum_{j \neq j'} \sum_{s} g_{i,j} g_{i',j'} \tau_s^2 u_{s,i} u_{s,i'} v_{s,j} v_{s,j'}}_{A_2}$$

$$+ \underbrace{\sum_{i \neq i'} \sum_{j} \sum_{s \neq s'} g_{i,j} g_{i',j} \tau_s \tau_{s'} u_{s,i} u_{s',i'} v_{s,j} v_{s',j}}_{A_3} + \underbrace{\sum_{i \neq i'} \sum_{j} \sum_{s} g_{i,j} g_{i',j} \tau_s^2 u_{s,i} u_{s,i'} v_{s,j} v_{s,j}}_{A_4}$$

$$+ \underbrace{\sum_{i} \sum_{j \neq j'} \sum_{s \neq s'} g_{i,j} g_{i,j'} \tau_s \tau_{s'} u_{s,i} u_{s',i} v_{s,j} v_{s',j'}}_{A_5} + \underbrace{\sum_{i} \sum_{j \neq j'} \sum_{s} g_{i,j} g_{i,j'} \tau_s^2 u_{s,i} u_{s,i} v_{s,j} v_{s,j'}}_{A_6}$$

$$+ \underbrace{\sum_{i} \sum_{j} \sum_{s \neq s'} g_{i,j}^2 \tau_s \tau_{s'} u_{s,i} u_{s',i} v_{s,j} v_{s',j}}_{A_7} + \underbrace{\sum_{i} \sum_{j} \sum_{s} g_{i,j}^2 \tau_s^2 u_{s,i}^2 v_{s,j}^2}_{A_8}.$$

$$B = \left\langle \sum_{s=1}^{r} \tau_s \cdot (u_s \circ v_s), \sum_{s=1}^{r} \tau_s \cdot (u_s \circ v_s) \right\rangle = \sum_{i} \sum_{j} \sum_{s,s'} \tau_s \tau_{s'} u_{s,i} u_{s',i} v_{s,j} v_{s',j}$$

$$= \underbrace{\sum_{i} \sum_{j} \sum_{s \neq s'} \tau_s \tau_{s'} u_{s,i} u_{s',i} v_{s,j} v_{s',j}}_{B_1} + \underbrace{\sum_{i} \sum_{j} \sum_{s} \tau_s^2 u_{s,i}^2 v_{s,j}^2}_{B_2}.$$

Similar to the way of computing expectations for the mean above, When there are cross terms like $u_{s,i}$ or $v_{s,j}$ in the product of $A_i$ and $B_j$, then $\mathbb{E}_{u,v}[A_i B_j] = 0$. Therefore, it is easy to check that $\mathbb{E}_{u,v}[A_1 B] = \mathbb{E}_{u,v}[A_2 B] = \mathbb{E}_{u,v}[A_3 B] = \mathbb{E}_{u,v}[A_4 B] = \mathbb{E}_{u,v}[A_5 B] = \mathbb{E}_{u,v}[A_6 B] = 0$ and we have $\mathbb{E}_{u,v}[AB] = \mathbb{E}_{u,v}[(A_7 + A_8)(B_1 + B_2)]$. Then we consider the cross terms on $\tau_s$. In $A_8 B_1$ and $A_7 B_2$, there exist the independent $\tau_s$ term, that is, $\mathbb{E}_{\tau}[A_8 B_1] = \mathbb{E}_{\tau}[A_7 B_2] = 0$, and the expectation of $AB$ is $\mathbb{E}_{\tau,u,v}[AB] = \mathbb{E}_{\tau,u,v}[A_7 B_1 + A_8 B_2]$. For the first term, we have:

$$\mathbb{E}[A_7 B_1] = \mathbb{E}\left[ 2 \sum_{i} \sum_{j} \sum_{s \neq s'} g_{i,j}^2 \tau_s^2 \tau_{s'}^2 u_{s,i}^2 u_{s',i}^2 v_{s,j}^2 v_{s',j}^2 \right] = 2 \sum_{i} \sum_{j} \sum_{s \neq s'} g_{i,j}^2 = 2r(r-1) \sum_{i} \sum_{j} g_{i,j}^2.$$

For the second term, we have:

$$\mathbb{E}[A_8 B_2] = \mathbb{E}\left[ \sum_{i,i'} \sum_{j,j'} \sum_{s,s'} g_{i,j}^2 \tau_s^2 \tau_{s'}^2 u_{s,i}^2 u_{s',i'}^2 v_{s,j}^2 v_{s',j'}^2 \right]$$

$$= \mathbb{E}\left[ \sum_{i \neq i'} \sum_{j \neq j'} \sum_{s \neq s'} g_{i,j}^2 \tau_s^2 \tau_{s'}^2 u_{s,i}^2 u_{s',i'}^2 v_{s,j}^2 v_{s',j'}^2 \right] + \mathbb{E}\left[ \sum_{i \neq i'} \sum_{j \neq j'} \sum_{s} g_{i,j}^2 \tau_s^4 u_{s,i}^2 u_{s,i'}^2 v_{s,j}^2 v_{s,j'}^2 \right]$$

$$+ \mathbb{E}\left[ \sum_{i \neq i'} \sum_{j} \sum_{s \neq s'} g_{i,j}^2 \tau_s^2 \tau_{s'}^2 u_{s,i}^2 u_{s',i'}^2 v_{s,j}^2 v_{s',j}^2 \right] + \mathbb{E}\left[ \sum_{i \neq i'} \sum_{j} \sum_{s} g_{i,j}^2 \tau_s^4 u_{s,i}^2 u_{s,i'}^2 v_{s,j}^4 \right]$$

$$+ \mathbb{E}\left[\sum_i \sum_{j \neq j'} \sum_{s \neq s'} g_{i,j}^2 \tau_s^2 \tau_{s'}^2 u_{s,i}^2 u_{s',i}^2 v_{s,j}^2 v_{s',j'}^2\right] + \mathbb{E}\left[\sum_i \sum_{j \neq j'} \sum_s g_{i,j}^2 \tau_s^4 u_{s,i}^4 v_{s,j}^2 v_{s,j'}^2\right]$$

$$+ \mathbb{E}\left[\sum_i \sum_j \sum_{s \neq s'} g_{i,j}^2 \tau_s^2 \tau_{s'}^2 u_{s,i}^2 u_{s',i}^2 v_{s,j}^2 v_{s',j}^2\right] + \mathbb{E}\left[\sum_i \sum_j \sum_s g_{i,j}^2 \tau_s^4 u_{s,i}^4 v_{s,j}^4\right]$$

$$= \sum_{i \neq i'} \sum_{j \neq j'} \sum_{s \neq s'} g_{i,j}^2 + \sum_{i \neq i'} \sum_{j \neq j'} \sum_s 3g_{i,j}^2 + \sum_{i \neq i'} \sum_j \sum_{s \neq s'} g_{i,j}^2 + \sum_{i \neq i'} \sum_j \sum_s 9g_{i,j}^2$$

$$+ \sum_i \sum_{j \neq j'} \sum_{s \neq s'} g_{i,j}^2 + \sum_i \sum_{j \neq j'} \sum_s 9g_{i,j}^2 + \sum_i \sum_j \sum_{s \neq s'} g_{i,j}^2 + \sum_i \sum_j \sum_s 27g_{i,j}^2$$

$$= \left(mnr^2 + 2mnr + 6mr + 6nr + 12r\right) \sum_i \sum_j g_{i,j}^2.$$

Thus, we can consolidate all the results as follows:

$$\mathbb{E}\|\frac{1}{r} \lim_{\rho \to 0} \nabla^0 f(w, \xi) - \nabla f(W, \xi)\|^2 = \frac{1}{r^2}\mathbb{E}\left[A \cdot B\right] - \|\nabla f(W, \xi)\|^2$$

$$= \frac{1}{r^2}\mathbb{E}\left[A_7 B_1 + A_8 B_2\right] - \|\nabla f(W, \xi)\|^2 = \left(1 + mn + \frac{2mn}{r} + \frac{6(m+n)}{r} + \frac{10}{r}\right)\|\nabla f(W, \xi)\|^2.$$

This completes the proofs.

### C.2 PROOFS OF THEOREM 2

We first introduce some basic lemmas for the subsequent proofs. In fact, when considering the properties of the function at each layer, we treat the parameters and gradients as a 2D matrices. However, to consider its general property, we treat them as a flattened parameter vector concatenation across layers. Therefore, in our proof, we slightly abuse both uppercase and lowercase letters, e.g., $\nabla f(Z)$ and $\nabla f(z)$, to express the specific properties of the gradient.

**Lemma 1** *Under Assumption 1 and 2, ZO gradient of* `TeZO` *is an unbiased estimator of the full FO gradient* $\nabla f(W)$ *with the variance:*

$$\mathbb{E}\|\frac{1}{r}\nabla^0 f(W, \xi) - \nabla f(W)\|^2 \leq \rho^2 \lambda^2 \delta_\rho + (\delta + 1)\sigma^2 + \delta\mathbb{E}\|\nabla f(W)\|^2, \quad (12)$$

*where* $\delta = 1 + mn + \frac{2mn}{r} + \frac{6(m+n)}{r} + \frac{10}{r}$ *and* $\delta_\rho = \frac{15r^2(m+3)^3(n+3)^3 + 36r^2(r-1)m^3n^3 + r^2(r-1)(r-2)m^3n^3}{4}$ *are two constants.*

**Proof.** *According to the studies of Nesterov & Spokoiny (2017); Chen et al. (2024); Yu et al. (2024), we first consider the smoothness property as follows:*

$$f(W + \rho Z, \xi) - f(W, \xi) - \langle\nabla f(W, \xi), \rho Z\rangle \leq \frac{\lambda}{2}\|\rho Z\|^2 = \frac{\rho^2 \lambda}{2}\|Z\|^2.$$

*Then we learn the distance between* $\nabla^0 f(W)$ *and the* $\lim_{\rho \to 0} \nabla^0 f(W)$. *Specifically, we consider the unbiased form as:*

$$\|\frac{1}{r}\nabla^0 f(W, \xi) - \frac{1}{r}\lim_{\rho \to 0}\nabla^0 f(W, \xi)\|^2$$

$$= \frac{1}{r^2}\|\frac{f(W + \rho Z, \xi) - f(W - \rho Z, \xi)}{2\rho} \cdot Z - \langle\nabla f(W, \xi), Z\rangle \cdot Z\|^2$$

$$= \frac{1}{r^2}\|\frac{f(W + \rho Z, \xi) - f(W, \xi) + f(W, \xi) - f(W - \rho Z, \xi) - 2\langle\nabla f(W, \xi), \rho z\rangle}{2\rho} \cdot Z\|^2$$

$$= \frac{1}{r^2}\left|\frac{(f(W + \rho Z, \xi) - f(W, \xi) - \langle\nabla f(W, \xi), \rho Z\rangle) - (f(W - \rho Z, \xi) - f(W, \xi) - \langle\nabla f(W, \xi), -\rho Z\rangle)}{2\rho}\right|^2 \cdot \|Z\|^2$$

$$\leq \frac{\rho^2 \lambda^2}{4r^2}\|Z\|^6.$$

*Substituting $Z = \sum_{s=1}^{r} \tau_s \cdot u_s \circ v_s$ and taking the expectation, we have:*

$$\mathbb{E}\|\frac{1}{r}\nabla^0 f(W,\xi) - \frac{1}{r}\lim_{\rho\to 0}\nabla^0 f(W,\xi)\|^2 \leq \frac{\rho^2\lambda^2}{4r^2}\mathbb{E}\|Z\|^6 = \frac{\rho^2\lambda^2}{4r^2}\mathbb{E}\|\sum_{s=1}^{r}\tau_s \cdot u_s \circ v_s\|^6$$

$$= \frac{\rho^2\lambda^2}{4r^2}\mathbb{E}\left(\|\sum_{s=1}^{r}\tau_s\cdot u_s\circ v_s\|^2\right)^3 \leq \frac{\rho^2\lambda^2}{4r^2}\mathbb{E}\left(r\sum_{s=1}^{r}\tau_s^2\|u_s\circ v_s\|^2\right)^3 = \frac{r\rho^2\lambda^2}{4}\mathbb{E}\left(\sum_{s=1}^{r}\tau_s^2\|u_s\|^2\|v_s\|^2\right)^3.$$

*Similarly, we can expand the term as:*

$$\mathbb{E}\left(\sum_{s=1}^{r}\tau_s^2\|u_s\|^2\|v_s\|^2\right)^3 = \mathbb{E}\sum_{s}\sum_{s'}\sum_{s''}\tau_s^2\tau_{s'}^2\tau_{s''}^2\|u_s\|^2\|u_{s'}\|^2\|u_{s''}\|^2\|v_s\|^2\|v_{s'}\|^2\|v_{s''}\|^2$$

$$= \mathbb{E}\sum_{s}\sum_{s'=s}\sum_{s''=s'}\tau_s^6\|u_s\|^6\|v_s\|^6 + \mathbb{E}\sum_{s}\sum_{s'=s}\sum_{s''\neq s'}\tau_s^4\tau_{s''}^2\|u_s\|^4\|u_{s''}\|^2\|v_s\|^4\|v_{s''}\|^2$$

$$+ \mathbb{E}\sum_{s}\sum_{s'\neq s}\sum_{s''=s}\tau_s^4\tau_{s'}^2\|u_s\|^4\|u_{s'}\|^2\|v_s\|^4\|v_{s'}\|^2 + \mathbb{E}\sum_{s}\sum_{s'\neq s}\sum_{s''=s'}\tau_s^2\tau_{s'}^4\|u_s\|^2\|u_{s'}\|^4\|v_s\|^2\|v_{s'}\|^4$$

$$+ \mathbb{E}\sum_{s}\sum_{s'\neq s}\sum_{s''\neq s,s'}\tau_s^2\tau_{s'}^2\tau_{s''}^2\|u_s\|^2\|u_{s'}\|^2\|u_{s''}\|^2\|v_s\|^2\|v_{s'}\|^2\|v_{s''}\|^2.$$

*Then we will discuss each term one by one. Actually, since $\tau, u, v$ are independent from each other, the expectation can be separated term by term. Since $u_s \sim \mathcal{N}(0, I_m)$, $v_s \sim \mathcal{N}(0, I_n)$ and $\tau_s \sim \mathcal{N}(0,1)$, we have: $\mathbb{E}\|u_s\|^2 = m$, $\mathbb{E}\|u_s\|^4 = m(2m-1) \leq 2m^2$, $\mathbb{E}\|u_s\|^6 = m(15 + 3(m-1) + (m-1)(m-2)) \leq (m+3)^3$, $\mathbb{E}\|v_s\|^2 = n$, $\mathbb{E}\|v_s\|^4 = n(2n-1) \leq 2n^2$, $\mathbb{E}\|v_s\|^6 = n(15 + 3(n-1) + (n-1)(n-2)) \leq (n+3)^3$, $\mathbb{E}\left[\tau_s^2\right] = 1$, $\mathbb{E}\left[\tau_s^4\right] = 3$ and $\mathbb{E}\left[\tau_s^6\right] = 15$. Therefore, we can provide the upper bound:*

$$\mathbb{E}\left(\sum_{s=1}^{r}\tau_s^2\|u_s\|^2\|v_s\|^2\right)^3 \leq 15r(m+3)^3(n+3)^3 + 36r^2m^3n^3 + r^3m^3n^3.$$

*Let $\delta_\rho = \frac{15r^2(m+3)^3(n+3)^3 + 36r^3m^3n^3 + r^4m^3n^3}{4}$, then we have:*

$$\mathbb{E}\|\frac{1}{r}\nabla^0 f(W,\xi) - \frac{1}{r}\lim_{\rho\to 0}\nabla^0 f(W,\xi)\|^2 \leq \frac{r\rho^2\lambda^2}{4}\mathbb{E}\left(\sum_{s=1}^{r}\tau_s^2\|u_s\|^2\|v_s\|^2\right)^3 \leq \rho^2\lambda^2\delta_\rho.$$

*Combining it with the variance in Theorem 1, we can finish the proofs.*

Then we can easily solve the convergence for `TeZO`. Similarly, without loss of generality, we still consider the 2D parameters. Let $\eta \leq \frac{1}{\lambda(\delta+1)}$ By expanding the smoothness inequality, we have:

$$\mathbb{E}_t[f(W_{t+1})] \leq f(W_t) + \mathbb{E}_t\langle\nabla f(W_t), W_{t+1} - W_t\rangle + \frac{\lambda}{2}\mathbb{E}_t\|W_{t+1} - W_t\|^2$$

$$= f(W_t) + \eta\mathbb{E}_t\langle\nabla f(W_t), -G_t\rangle + \frac{\lambda\eta^2}{2}\mathbb{E}_t\|G_t\|^2$$

$$= f(W_t) - \eta\mathbb{E}_t\|\nabla f(W_t)\|^2 + \frac{\lambda\eta^2}{2}\mathbb{E}_t\|G_t\|^2$$

$$\leq f(W_t) - \eta\mathbb{E}_t\|\nabla f(W_t)\|^2 + \frac{\lambda\eta^2}{2}\mathbb{E}_t\|\frac{1}{r}\nabla^0 f(W_t,\xi) - \nabla f(W_t)\|^2 + \frac{\lambda\eta^2}{2}\mathbb{E}_t\|\nabla f(W_t)\|^2$$

$$\leq f(W_t) - \eta\left(1 - \frac{\lambda(1+\delta)\eta}{2}\right)\mathbb{E}_t\|\nabla f(W_t)\|^2 + \eta^2\rho^2\frac{\lambda^3\delta_\rho}{2} + \eta^2\frac{\lambda(\delta+1)\sigma^2}{2}$$

$$\leq f(W_t) - \frac{\eta}{2}\mathbb{E}_t\|\nabla f(W_t)\|^2 + \eta^2\rho^2\frac{\lambda^3\delta_\rho}{2} + \eta^2\frac{\lambda(\delta+1)\sigma^2}{2}.$$

Therefore, let $D_0 = f(W_0) - f(W_\star)$ be the initialized bias where $W_\star$ is the optimal solution, by accumulating it from $t = 0$ to $T - 1$ and taking the full expectation, we have:

$$\frac{1}{T}\sum_{t=0}^{T-1}\mathbb{E}\|\nabla f(W_t)\|^2 \leq \frac{2D_0}{\eta T} + \eta\lambda\left(\rho^2\lambda^2\delta_\rho + (\delta+1)\sigma^2\right).$$

By simply selecting the learning rate $\eta = \mathcal{O}\left(\sqrt{\frac{D_0}{\lambda T(\rho^2 \lambda^2 \delta_\rho + \delta \sigma^2)}}\right) \leq \frac{1}{\lambda(\delta+1)}$, we have:

$$\frac{1}{T} \sum_{t=0}^{T-1} \mathbb{E}\|\nabla f(W_t)\|^2 = \mathcal{O}\left(\sqrt{\frac{\lambda D_0 \left(\rho^2 \lambda^2 \delta_\rho + \delta \sigma^2\right)}{T}}\right).$$

