# OpenReview forum: "TeZO: Empowering the Low-Rankness on the Temporal Dimension in the Zeroth-Order Optimization for Fine-tuning LLMs"
_ICLR.cc/2026/Conference — Submitted to ICLR 2026_

### Official Review · Reviewer_yUwe · 2025-10-29

**Soundness:** 3
**Presentation:** 3
**Contribution:** 3
**Rating:** 8
**Confidence:** 4

**Summary:**

This paper proposes TeZO, a Zeroth-Order fine-tuning method that captures both model-wise and temporal low-rank structures using Canonical Polyadic Decomposition. By jointly factorizing perturbations across iterations, TeZO-Adam reduces optimizer-state memory from O(d) to O(r)—requiring only ~35% of MeZO-Adam’s memory—while maintaining comparable accuracy. The method provides theoretical guarantees of unbiasedness and convergence and demonstrates strong efficiency on large LLMs (OPT-13B, LLaMA-7B). Overall, TeZO enables Adam-level adaptivity with SGD-level memory cost, advancing scalable fine-tuning for large models.

**Strengths:**

1. Innovative temporal low-rank design:

Proposes the first Zeroth-Order (ZO) fine-tuning framework that captures temporal low-rankness across training iterations via Canonical Polyadic Decomposition (CPD), extending previous gradient-only low-rank models (LOZO, SubZO).

2. Theoretical soundness:

Provides rigorous proofs (Theorems 1–2) demonstrating that TeZO is an unbiased estimator with convergence and variance guarantees equivalent to first-order optimizers, while maintaining lower sampling complexity.

3. Dynamic rank selection mechanism:

Introduces adaptive, layer-wise rank estimation (Eq. 5–6) derived from parameter rank propagation, improving stability, avoiding manual tuning, and yielding an average rank ≈ 31.5 (Appendix B.1.4, Table 5).

4. Superior memory efficiency:

TeZO-Adam reuses shared CPD factors (U, V, χ) for both first- and second-order moments, reducing optimizer-state storage from O(d) to O(r); achieves ~65 % memory savings versus MeZO-Adam and ~30 % versus MeZO-M (Table 2 and Figure 4a).

5. Computational efficiency:

Maintains 1.63× faster wall-clock time than MeZO-Adam (Figure 4b) by avoiding redundant state expansion; the separable second-moment approximation (Eq. 8) effectively suppresses cross-term overhead.

6. Extensibility across optimizers:

Supports TeZO-SGD, TeZO-Momentum, TeZO-Adam, and TeZO-LION variants with consistent accuracy and stability across architectures (OPT-13B, LLaMA-7B/30B) and datasets (SST-2, BoolQ, WIC, RTE).

7. Strong empirical validation:

Evaluations on 16 tasks confirm state-of-the-art memory–efficiency trade-off without accuracy loss, establishing TeZO as a practical solution for LLM fine-tuning under limited GPU memory.

8. Practical relevance:

Offers a feasible fine-tuning paradigm for large models (> 3 B parameters) on single-device environments, addressing real-world memory bottlenecks in LLM adaptation.

**Weaknesses:**

1. Unclear explanation of TeZO-Adam’s superior memory benefit:

The paper does not explicitly analyze why TeZO-Adam outperforms other TeZO variants. The gain originates from reusing CPD factor vectors for both momentum orders—removing the need to store full optimizer states—but this is only briefly mentioned. A more detailed breakdown of memory allocation among weights, first-, and second-order states is needed.

2. Accuracy–efficiency trade-off imbalance:

Improvements are dominated by computational and memory efficiency; accuracy gains are marginal (< 0.3 %). The method preserves rather than enhances downstream performance.

3. Incomplete quantification of CPD overhead:

Although TeZO-Adam is reported as 1.63× faster, the runtime cost of CPD factor updates, decomposition, and sampling is not isolated. A per-component runtime profile would clarify whether the speedup derives from algorithmic design or implementation details.

4. Restricted scalability window:

Appendix B.4 indicates TeZO’s benefits emerge mainly for models ≥ 3 B parameters; for smaller models (< 1 B), the CPD overhead offsets gains. Clarifying this crossover point quantitatively (e.g., “break-even” size) is necessary.

5. Assumption fragility of temporal low-rankness:

The method assumes gradient subspaces remain stable across iterations. Gradient similarity maps (Fig. 6) show partial deviations; robustness under domain shift, curriculum, or non-stationary fine-tuning remains untested.

6. Limited hyperparameter sensitivity study:

Critical parameters (ρ, rₜₕ, rₘₐₓ) are fixed empirically; no systematic exploration of their impact on convergence, stability, and efficiency.

7. Dynamic rank generalization scope:

Layer-wise rank adaptation was evaluated only on OPT-13B; no evidence is given for generality on LLaMA or RoBERTa.

8. Variance and convergence not empirically validated:

Although variance constant δ > MeZO’s is theoretically derived, its empirical influence on convergence rate or oscillation is unexamined.

9. Numerical stability concerns:

Reusing factor vectors χ over T steps may cause rounding errors or rank collapse; the paper omits orthogonalization or normalization countermeasures.

10. Limited discussion of optimizer comparison:

TeZO-Adam’s gains stem from storage compression, not from improved optimization dynamics; no analysis of curvature adaptation or learning-rate behavior relative to TeZO-M or TeZO-SGD.

11. Weak integration with PEFT methods:

Although “ZO + LoRA” is briefly mentioned, the claim of negligible benefit lacks quantitative evidence. Exploring complementary hybridization could further enhance applicability.

12. Insufficient visualization of contribution hierarchy:

The paper could better emphasize that TeZO-Adam achieves Adam-level adaptivity with SGD-level memory through a schematic of optimizer-state composition (MeZO vs. LOZO vs. TeZO).

13. Scant analysis of temporal rank drift:

Temporal subspace stability over long training (e.g., 20 K steps) is not monitored. Reporting cosine similarity decay or principal-angle changes between gradient spaces would clarify robustness.

**Questions:**

Can the authors provide a detailed memory-component table showing how TeZO-Adam reuses CPD factors (U, V, χ) to compress both first- and second-moment states?

What is the precise runtime share of CPD updates versus forward passes and factor sampling across different model scales?

At what parameter size does TeZO-Adam begin to outperform MeZO-Adam in both memory and time? Can a “break-even analysis” be added?

How does TeZO-Adam behave when gradient subspaces shift—e.g., under domain adaptation or curriculum learning? Could adaptive temporal rank re-estimation mitigate performance drop?

Theoretical δ is slightly higher than MeZO’s—can authors empirically show variance trends or convergence oscillations over training iterations?

 Beyond layer-wise adaptation, could rank r be adjusted over time based on gradient variance or loss curvature?

 How do changes in ρ, rₜₕ, and rₘₐₓ affect memory usage, convergence, and accuracy? A sensitivity plot would enhance reproducibility.

Optimizer-variant comparison: Could the authors include learning-dynamics plots (e.g., effective learning-rate curves) to distinguish TeZO-Adam’s adaptive behavior from TeZO-M or SGD?

[If the paper gets accepted for publication, I \recommend that the authors publicly release their code to further enhance its impact and visibility.]

---

### Official Review · Reviewer_Q36h · 2025-10-29

**Soundness:** 2
**Presentation:** 3
**Contribution:** 3
**Rating:** 4
**Confidence:** 5

**Summary:**

This paper introduces a novel low-rank zeroth-order optimizer, TeZO, which simultaneously considers the low-rank structures in both the model and temporal dimensions. Specifically, canonical polyadic decomposition (CPD) is employed to represent the zeroth-order perturbations as a sum of three-dimensional tensors, thereby significantly reducing the computational and storage costs. Moreover, a dynamic rank selection mechanism is integrated at each layer to render the low-rank representation more adaptive. To further enhance the practical utility of the optimizer, memory-efficient momentum variant TeZO-m and adaptive variant TeZO-Adam are proposed, with TeZO-Adam consuming only approximately 35% of the memory of MeZO-Adam. Theoretical analyses demonstrate that TeZO provides an unbiased estimation of the first-order gradient, exhibiting comparable variance and convergence rates to existing zeroth-order methods. Extensive experimental results further validate the efficacy of TeZO in fine-tuning large language models. The primary innovation of this approach lies in extending the low-rank property to the temporal dimension, thereby achieving a more efficient optimization process.

**Strengths:**

The paper is well-written, with clear and rigorous exposition. Its central claims---including gradient low-rankness, efficient perturbation generation, provable convergence, and superior empirical performance---are strongly supported by both theoretical analysis and extensive experiments.

The proposed TeZO method represents the sequence of ZO perturbation matrices as a third-order tensor $\mathcal{T} \in \mathbb{R}^{T \times m \times n}$ and applies Canonical Polyadic Decomposition (CPD) to approximate it via a sum of rank-one factors, significantly reducing memory and computational cost.

Theoretical analysis establishes a standard convergence rate under typical settings of learning rate $\eta$ and smoothing parameter $\mu$.

Experiments follow conventional ZO fine-tuning benchmarks (e.g., RoBERTa-Large on GLUE) and are extended to large-scale settings (e.g., LLaMA), demonstrating the method's practicality and scalability.

**Weaknesses:**

1) The role of the temporal dimension remains ambiguous. The authors introduce low-rankness across training steps but fail to justify why this is beneficial, especially compared to existing methods.

2) Unlike other low-rank ZO methods (e.g., LoZO, SubZero) that update perturbation bases periodically, this work uses fixed feature vectors throughout training. The authors did not adequately explain why a static basis performs well, undermining the novelty and plausibility of the approach.

I will re-score based on the author's rebuttal.

**Questions:**

see the Weaknesses.

**Details Of Ethics Concerns:**

None.

---

### Official Review · Reviewer_TFYx · 2025-10-30

**Soundness:** 3
**Presentation:** 2
**Contribution:** 2
**Rating:** 2
**Confidence:** 4

**Summary:**

This paper proposes TeZO, a low-rank ZO optimization method that introduces per iteration and temporal low-rankness of gradient to improve memory efficiency and training speed in fine-tuning LLMs. The method adaptively selects rank per layer and can be extended to momentum and Adam variants. Experiments on mutiple LLMs show TeZO achieves better memory efficiency with comparable or slightly better accuracy.

**Strengths:**

Strengths

* The key idea of exploiting low-rank structure not only across model dimensions but also along the temporal axis is well-motivated.
* TeZO achieves memory savings (35% of MeZO-Adam) and wall-clock speedup (1.63× faster than MeZO-Adam)

**Weaknesses:**

Weaknesses and Question:

* The motivation seems comfusing, the observation is based on the gradient of first-order fine-tuning has per-iter and temporal low-rankness. But the paper is target on ZO fine-tuning, so the observation supposed to based on gradient of ZO, or should claim clearly the relationship betwween gradient of ZO and FO.
* Strongly correlated with the previous point, it's true that there is low-rankness in each FO step. However, for a ZO step with one perturbation (I assume its the scenario this paper focus on from Algorithm 2), the gradient is computed by a loss difference and a normal distributed perturbation, so the gradient seems not supposed to be low-rankness. Could you correct me or provide more details?
* In the MeZO paper, they claim MeZO with Adam, performance will not be significantly improved, which is different from the experiment results in this paper. So in which settings will performance be better with Adam? If the performance improvement after adding Adam is not significant, then there is no need to use mezo-adam.
* Without Adam, Tezo performs worse than Mezo.
* Could you provide a breakdown of the wall-clock time for calculating Mezo and Tezo over tasks with longer sequence , since forward operations would likely consume most of the time in this case? This would allow us to determine the speed improvement of Tezo under less-than-ideal conditions (e.g., small model but long sequence).
* Aside from wall-clock time, is there an training iteration acceleration? As other low-rank method claim to be converged faster.
* It would be better to provide more fine-tuning results on more challenging tasks, on MMLU or MT-Bench, like the setting in [1].

[1] Lisa: Layerwise importance sampling for memory-efficient large language model fine-tuning

**Questions:**

Please refer to the Weakness part.

---

### Official Review · Reviewer_FbdW · 2025-11-08

**Soundness:** 2
**Presentation:** 2
**Contribution:** 1
**Rating:** 2
**Confidence:** 4

**Summary:**

This paper addresses the high memory cost of stateful optimizers, such as Adam, in Zeroth-Order (ZO) fine-tuning for LLMs. The authors posit that gradients are not only low-rank individually but also share a similar low-rank subspace across the temporal dimension. Leveraging this, the proposed TeZO method employs Canonical Polyadic Decomposition (CPD) to compress optimizer states from $\mathcal{O}(d)$ to $\mathcal{O}(r)$. Experiments demonstrate that TeZO-Adam achieves state-of-the-art performance while using only about 35% of the memory of MeZO-Adam, even less than the standard MeZO-SGD.

**Strengths:**

1. The paper is well-written, and the proposed idea is easy to follow.

2. The method demonstrates compelling empirical efficiency, reporting significant memory savings that enable the use of Adam-like optimizers with less overhead than even standard ZO-SGD.

**Weaknesses:**

1. **On the Necessity and Validity of Stateful Optimizers in ZO**: The paper's core motivation is that the memory cost of optimizer states prevents the use of stateful optimizers (e.g., Adam) in existing low-rank ZO methods.
	* Questionable Premise: This motivation hinges on the assumption that stateful optimizers are necessary for strong ZO performance. This is not a settled issue. Prior literature (e.g., MeZO, ZO-Bench [1]) has often reported that Adam provides marginal or no benefit over well-tuned SGD in the ZO context.
	* Questionable Mechanism: The implementation in Algorithm 2 raises questions about whether the "momentum" and "variance" terms are functioning as they do in first-order optimization. In FO, $M_t$ tracks a meaningful gradient direction. In TeZO, $M_t$ appears to accumulate scaled random variables ($\kappa\tau$). It is unclear if this EMA of random vectors, which are uncorrelated with the true gradient, provides meaningful guidance.
	* Contradictory Results and Tuning Bias: The paper does show that its Adam variants outperform SGD variants (Tables 3 & 4), contradicting some previous findings. This discrepancy requires careful analysis. A potential confounding factor is visible in Table 7 (Hyperparameter configurations). The SGD variants (e.g., MeZO) were tuned over a very coarse, log-scale grid of learning rates (e.g., {1e-4, 1e-5, 1e-6, 1e-7}). The Adam variants, in contrast, were tuned over a much finer-grained grid (e.g., {1e-4, 3e-5, 1e-5, 3e-6}). This suggests the reported performance gap may be an artifact of tuning bias rather than an intrinsic algorithmic advantage. It seems essential to re-run SGD variants with the same fine-grained tuning to ensure a fair comparison.

2. **Incremental Methodological Novelty and Design Questions**: The core method, while effective, appears to be a highly incremental extension of prior work.
	* LoZO and Subzero already introduced the concept of low-rank decomposition for ZO. TeZO's contribution is to extend this from 2D (spatial) to 3D (temporal), which seems to be a straightforward modification.
	* Furthermore, the design choice for this temporal extension is counterintuitive. The method claims to "empower low-rankness on the temporal dimension" yet does so by fixing the $u_s, v_s$ subspace after random initialization. One might argue that a method like LOZO, which periodically updates the subspace, is a more logical approach to handling temporal dynamics (i.e., subspace drift) than TeZO's fixed-subspace approach.

3. **Outdated Experimental Setup**
	* To validate the practical utility of ZO methods in 2025, a more demanding experimental setup seems necessary. The models used (OPT, LLaMA) and tasks (mostly GLUE-style benchmarks) are becoming outdated. The field of ZO for LLMs has matured, and its true value must be demonstrated on current-generation models (e.g., Llama-3, Gemma-3) and on more complex, challenging tasks (e.g., code, math, instruction tuning) where fine-tuning is most impactful.

4. **Lack of a Discussion about Limitations**
	* The paper does not include dedicated discussions on limitations. This omission prevents a clear understanding of the authors' own assessment of their method's potential failure modes, sensitivities (e.g., to the fixed random $u_s, v_s$), or boundaries of applicability. A frank discussion of these limitations would strengthen the paper's contribution.


[1] Zhang, Yihua, et al. "Revisiting Zeroth-Order Optimization for Memory-Efficient LLM Fine-Tuning: A Benchmark." International Conference on Machine Learning. PMLR, 2024.

**Questions:**

See weaknesses

---

### Meta-Review · Area_Chair_mMoM · 2026-01-05

**Summary:**

This paper proposes TeZO, a novel zeroth-order (ZO) optimization method designed for fine-tuning Large Language Models (LLMs). The core contribution lies in exploiting the low-rankness of gradients not only within individual layers but also across the temporal dimension (training steps). By representing ZO perturbations as a 3D tensor and employing Canonical Polyadic Decomposition (CPD), the authors aim to compress optimizer states—particularly for memory-intensive optimizers like Adam—thereby reducing memory consumption while attempting to maintain performance.

While the reviewers appreciated the motivation to improve memory efficiency in ZO fine-tuning, several critical concerns were raised regarding the paper's soundness and experimental rigor. Reviewers questioned the fundamental validity of the proposed "momentum" mechanism, noting that accumulating scaled random variables might not provide meaningful guidance in a ZO context. Significant concerns were also raised regarding experimental fairness; specifically, there appears to be a "tuning bias" where the proposed Adam variants were tuned on a fine-grained grid while baselines were tuned on a coarse grid. Furthermore, reviewers pointed out that the experimental setup, which relies on older models (e.g., OPT, LLaMA-1) and GLUE-style benchmarks, is becoming outdated for evaluating modern LLM fine-tuning strategies. Questions regarding the counterintuitive design of using a fixed subspace rather than updating it periodically also remain unanswered.

Unfortunately, the authors did not submit a rebuttal to address these substantial critiques. Consequently, the reviewers' concerns regarding the algorithmic justification, unfair comparisons, and experimental relevance remain entirely unresolved. Given these outstanding issues and the lack of author response, the paper cannot be recommended for acceptance.

**Reviewer Concerns:**

Since the authors did not submit a rebuttal, all the reviewer concerns remain entirely unresolved.

**Reviewer Scores:**

Since the authors did not submit a rebuttal, all the reviewer concerns remain entirely unresolved.

---

### Decision · Program_Chairs · 2026-01-26

Reject